

# A method for measuring total aerosol oxidative potential (OP) with the dithiothreitol (DTT) assay and comparisons between an urban and roadside site of water-soluble and total OP

Dong Gao[1], Ting Fang[2], Vishal Verma[3], Linghan Zeng[2], Rodney J. Weber[2]

[1]School of Civil and Environmental Engineering, Georgia Institute of Technology, Atlanta, GA 30332, USA
[2]School of Earth and Atmospheric Sciences, Georgia Institute of Technology, Atlanta, GA 30332, USA
[3]School of Civil and Environmental Engineering, University of Illinois at Urbana-Champaign, Urbana, IL 61801, USA

*Correspondence to*: Rodney J. Weber (rweber@eas.gatech.edu)

**Abstract**. An automated analytical system was developed for measuring the oxidative potential (OP) with the dithiothreitol (DTT) assay of filter extracts that include both water-soluble and water-insoluble (solid) aerosol species. Three approaches for measuring total oxidative potential were compared. These include using methanol as the solvent with (1) and without (2) filtering the extract, followed by removing the solvent and reconstituting with water, and (3) extraction in pure water and performing the OP analysis in the extraction vial with the filter. The water extraction method (the third approach, with filter remaining in the vial) generally yielded highest DTT responses with better precision (coefficient of variation of 1−5 %), and was correlated with a greater number of other aerosol components. Because no organic solvents were used, which must be mostly eliminated prior to DTT analysis, it was the easiest to automate by modifying an automated analytical system for measuring water-soluble OP developed by Fang et al. (2015). Daily 23h filter samples were collected simultaneously at a roadside (RS) and a representative urban (GT) site for two one-month study periods, and both water-soluble (OP$^{WS-DTT}$) and total (OP$^{Total-DTT}$) OP were measured. Using PM$_{2.5}$ (aerodynamic diameter < 2.5 $\mu$m) high-volume samplers with quartz filters, the OP$^{WS-DTT}$ to OP$^{Total-DTT}$ ratio at the urban site was 65 % with a correlation coefficient (r) of 0.71 (N=35; p-value<0.01), compared to a ratio of 62 % and r=0.56 (N=31; p-value<0.01) at the roadside site. Similar results were found using particle composition monitors (flow rate of 16.7 L min$^{-1}$) with Teflon filters. Comparison of measurements between sites showed only slightly higher levels of both OP$^{WS-DTT}$ and OP$^{Total-DTT}$ at the RS site, indicating both OP$^{WS-DTT}$ and OP$^{Total-DTT}$ were largely spatially homogeneous. These results are consistent with roadway emissions as sources of DTT-quantified PM$_{2.5}$ OP and that both soluble and insoluble aerosol components contributing to OP are largely secondary.

## 1. Introduction

Exposure to ambient particulate matter (PM) is associated with adverse health effects (Atkinson et al., 2001; Li et al., 2003a; Lim et al., 2013; Pope, 1995; Pope and Dockery, 2006). The mechanisms of PM toxicity are complex and not completely understood. One view is that PM toxicity occurs through inducement of oxidative stress (Delfino et al., 2005; Delfino et al., 2013; Nel, 2005); a state of biochemical imbalance in which the presence and formation of reactive oxygen species (ROS) in



the human body overwhelms antioxidant defenses, eventually leading to various adverse health outcomes (Delfino et al., 2011; Donaldson et al., 2001; Li et al., 2003a). ROS can be either transported on inhaled particles to the air–lung interface or generated in vivo by interaction between deposited PM and physiological chemical components (Lakey et al., 2016). The ability of PM to generate ROS is defined as the oxidative potential (OP) of PM. OP integrates various biologically relevant

properties of particles, including size, surface and chemical composition, which may better reflect the biological response to PM exposure and consequently be more informative than PM mass, or specific PM chemical species, when attempting to link aerosols to adverse health effects.

Various methods have been developed to assess PM OP (Ayres et al., 2008; Cho et al., 2005; Mudway et al., 2004; Shi et al., 2003). The dithiothreitol (DTT) assay is used in this study to measure the OP of fine particles (i.e., $OP^{DTT}$). DTT acts as a

surrogate of cellular reductants, such as NADH/NADPH. The goal is to mimic interactions between physiological reductants and aerosol components through a purely chemical analysis. Various aerosol components can react directly with antioxidants (reducing agents), or transfer electrons from the antioxidants to dissolved oxygen, leading to antioxidant depletion in the first case and both antioxidant depletion and ROS generation in the second. In the DTT assay, physiological reductants are represented by DTT. When this reaction is monitored under conditions of excess DTT, the DTT consumption over time is

proportional to the concentration of PM redox-active species, quantified as $OP^{DTT}$. $OP^{DTT}$ per volume of air sampled has been found to correlate with biological markers, such as cellular hemeoxygenase (HO-1) expression (Li et al., 2003b) and fractional exhaled nitric oxide ($FE_{NO}$) in human subjects (Delfino et al., 2013; Janssen et al., 2015). Epidemiological studies have linked $OP^{DTT}$ to adverse health outcomes, such as asthma, rhinitis (Yang et al., 2016) and asthma or wheezing and congestive heart failure (Bates et al., 2015; Fang et al., 2016). Utilizing different measures of OP (e.g., ascorbic acid, AA;

glutathione, GSH; uric acid, UA), some other studies, however, have not found links between OP and adverse health effects (Atkinson et al., 2016; Canova et al., 2014).

$OP^{DTT}$ of water-soluble PM components (referred to as $OP^{WS-DTT}$) is the common focus of OP studies since it is the most straightforward to measure. Researchers have identified DTT-active water-soluble PM components, including HUmic-Like Substances (HULIS) (Lin and Yu, 2011; Verma et al., 2012; Verma et al., 2015b), oxygenated quinones (a subset of HULIS)

(Cho et al., 2005; Kumagai et al., 2002), and transition metals (Charrier and Anastasio, 2012; Fang et al., 2016; Verma et al., 2015a). Water-insoluble species can also be an important fraction of the overall PM redox activity. Li et al. (2013) found that the solid particle phase was a dominant factor in the DTT-based redox activity of soot particles. Akhtar et al. (2010) found that redox-active substances could be strongly bound to solid particles and not be easily extracted by water. McWhinney et al. (2013) reported that 89 %–99 % of the redox activity of diesel exhaust particles (DEP) were water-insoluble and not

extractable by moderately polar (methanol) and nonpolar (dichloromethane) organic solvents. Daher et al. (2011) reported the highest intrinsic $OP^{DTT}$ for particle collection with a Biosampler, which was considered most efficient in capturing both the soluble and insoluble PM species. Including the contribution of water-insoluble species in the OP assessment would be



closer to actual PM exposure. A measure of both water-soluble and water-insoluble OP would be useful to elucidate the relative risks of water-soluble versus water-insoluble OP-induced health risks for specific health endpoints, such as

respiratory versus cardiovascular dysfunction.

Several PM extraction methods have been used to assess the OP of water-insoluble PM. A common approach is to extract water-insoluble species in organic solvents, such as methanol and dichloromethane. Verma et al. (2012) found $OP^{DTT}$ (expressed per µg of PM mass) of filtered methanol extracts to be correlated with water-insoluble organic carbon and elemental carbon (N=8). The DTT assay response for the methanol extracts was significantly higher than that for the water

extracts with methanol-to-water $OP^{DTT}$ ratio of 1.6 ± 0.4. Yang et al. (2014) compared $OP^{DTT}$ of ambient PM with two extraction methods for Teflon filters: methanol extraction without filtering and water extraction. They found that the methanol extracts were more DTT-reactive (expressed per $m^3$ of sampled volume) than the water extracts. In this method, removal of organic solvent by evaporation was necessary prior to the DTT assay, which can result in the loss of labile redox-active PM species, such as semi-volatile organic compounds.  Instead of attempting to dissolve water-insoluble species in

various solvents, other studies perform the assay in the extraction liquid without filtration, retaining the insoluble particles in the DTT reaction solution. McWhinney et al. (2013) measured total redox activity of DEP using particle suspensions that were obtained by a water-extraction procedure, after which the filter was removed from the analysis. Whereas Charrier et al. (2016) performed the DTT assay on the extraction liquid that still contained the filter. Daher et al. (2011) collected particles directly into water with a BioSampler and performed the DTT analysis without filtration.

In this study, we assess techniques for quantifying the overall oxidative potential of ambient particles and determine the relative contribution from water-soluble and water-insoluble components to PM OP by contrasting measurements from different sites. This was accomplished by conducting the DTT assay on samples extracted by three different methods. The goal was to develop a system for measuring both soluble and total $OP^{DTT}$ (insoluble $OP^{DTT}$ by difference) fractions to allow studies on the health effects of soluble (Bates et al., 2015) versus insoluble PM OP.

## 2. Experimental methods

### 2.1 Sampling methods and locations

Measurements were made at two contrasting sampling sites: Georgia Tech and Roadside. The Georgia Tech (GT) site was situated on the rooftop of the Ford ES&T building on the campus of Georgia Tech about 30 m above ground level and approximately 420 m from the roadside site. (A map of the sites is provided in Supplement Fig. S1.) The GT site is assumed

to be representative of the urban Atlanta environment. The Roadside (RS) site is adjacent (within 3 m) to a heavily trafficked interstate freeway (I-85/75) with an annual average daily traffic count of 382,000 vehicles in 2015 (Georgia Department of



Transportation (GDOT) traffic data, station ID 1215482). Heavy-duty trucks are restricted, resulting in predominantly light-duty gasoline vehicle traffic (non-heavy duty truck traffic nominally 95 %, GDOT 2013 data).

Measurements were undertaken during two different periods using different particle filter collection systems. A high-volume (HiVol) sampler (Thermo Anderson, nominal flow rate 1.13 $m^3$ $min^{-1}$, $PM_{2.5}$ impactor) was set up at each site to collect ambient fine particles simultaneously from 21 April 2016 to 30 May 2016. Fine particles were collected with prebaked 8×10 in. quartz filters (Pallflex Tissuquartz, Pall Life Sciences) for 23 h (11:00 am–10:00 am the next day). The HiVol quartz filters were wrapped in prebaked aluminum foil immediately after collection and stored at -18 ℃ until analyses. To avoid biases between the two HiVol samplers, they were operated side-by-side at GT for 9 days. Both water-

soluble $OP^{DTT}$ ($OP^{WS-DTT}$) and total $OP^{DTT}$ ($OP^{Total-DTT}$) (obtained by method 3, described below) were within 10 % (Supplement Fig. S2). In Sect. 3.4, OP data from HiVol 1 were adjusted to match HiVol 2 based on the orthogonal linear regression from this comparison. The factors used to convert OP from HiVol 1 to HiVol 2 were 1.00 and 1.10 for $OP^{WS-DTT}$ and $OP^{Total-DTT}$, respectively.

Zefluor PTFE membrane filters (diameter 47 mm, 2 $\mu$m pore size, Pall Life Sciences) were used as well for simultaneous

$PM_{2.5}$ sample collection from 26 July 2016 to 21 August 2016 using particle composition monitors (PCM, 16.7 L $min^{-1}$, $PM_{2.5}$ URG cyclone, un-denuded). Two PCMs were installed at each site to obtain two Teflon samples, one was used for $OP^{WS-DTT}$ and the other for $OP^{Total-DTT}$ analysis. Similar to the HiVol filter sampling, after 23 h collection, the PCM Teflon filters were placed into Petri dishes and stored at -18 ℃.

### 2.2 Measurements of PM oxidative potential

OP analyses were performed on both HiVol quartz and PCM Teflon filters. The DTT assay followed the protocol developed by Cho et al. (2005). All OP analyses on HiVol quartz filters were done immediately after collection; OP measurements on Teflon filters were completed within one month after collection.

### 2.2.1 $OP^{WS-DTT}$ analysis

One circular punch (diameter of 1 in.) from the HiVol quartz filter was extracted in 4.9 mL of deionized water (DI, >18 MΩ

$cm^{-1}$) in a sterile polypropylene centrifuge tube (VWR International LLC, Suwanee, GA, USA) via 30-min sonication. Experiments in using sonication versus shaking showed little difference indicating no bias due to possible radical formation during the sonication process (Miljevic et al., 2014), see Supplementary Material. The extract was then filtered through 0.45 $\mu$m PTFE syringe filters (Fisherbrand, Fisher Scientific) to remove insoluble material. The filtered PM water-extract was analyzed using a semi-automated system ($OP^{WS-DTT}$ system) developed by Fang et al. (2015) where all chemical reagents and

reaction mixtures were mixed and transferred by two programmable syringe pumps. Briefly, 3.5 mL water extract is



incubated with 0.5 mL of 1 mM DTT and 1 mL potassium phosphate buffer (K-buffer; pH=7.4) in a single incubation vial (IV) at 37 °C. At designated time points (0, 4, 13, 23, 32, 41 min), an aliquot (100 $\mu$L) of this mixture is transferred to another vial (reaction vial, RV) and mixed with trichloroacetic acid (TCA) to quench the reaction. Tris buffer (pH=8.9) and 5,5'-dithiobis-(2-nitrobenzoic acid) (DTNB) are then added to form a colored product which absorbs light at 412 nm. The final mixture is pushed through a 10cm path length liquid waveguide capillary cell (LWCC), and the absorbance at 412 nm is detected and recorded by an online UV–visible spectrophotometer. The DTT concentration at each time point is quantified based on the absorbance calibration curve, which had previously been determined from standard DTT solutions also containing TCA, Tris buffer, and DTNB. The DTT consumption rates are then determined by applying linear regression to the observed DTT concentration versus time. The final OP results are calculated by subtracting a blank value from the sample and normalized by the volume of air that passed through the filter (of 1-inch diameter punch size), expressed as nmol DTT min$^{-1}$ per sampled air volume (OP$^{WS\text{-}DTT}$ m$^{-3}$; if not explicitly stated, OP$^{WS\text{-}DTT}$ is OP$^{WS\text{-}DTT}$ m$^{-3}$) to provide a measure of atmospheric levels of water-soluble aerosol OP. The DTT consumption rate of multiple blanks for quartz filters (N=42) was stable with a mean ± 1$\sigma$ of 0.33 ± 0.07 nmol min$^{-1}$. 9,10-phenanthrenequinone (PQN) is used as positive control throughout the analysis to evaluate the stability of the analytical system.

Water extraction was also performed on the PCM Teflon filters. Each of the two Teflon filters collected simultaneously at each site was cut in half. One half of each filter was combined and immersed in 4.9 mL DI in a beaker and sonicated for 30 minutes. The water extract was then filtered and OP$^{WS\text{-}DTT}$ determined using the automated system. DTT analytical processing was exactly the same as that for quartz filters described above. The other filter halves were stored in a freezer until OP$^{Total\text{-}DTT}$ analysis. This analysis approach removed any potential biases associated with the separate filter collection systems at each site. Sample flow rates were measured at the beginning and end of sampling for each filter system and the overall average was used to calculate OP$^{WS\text{-}DTT}$ m$^{-3}$. Field blanks were also tested in the same manner and had an average slope plus or minus standard deviation of 0.35 ± 0.08 nmol min$^{-1}$ (mean ± 1$\sigma$, N=18).

### 2.2.2 OP$^{Total\text{-}DTT}$ analysis

***Sample extraction and preparation:*** To assess methods for characterizing OP$^{Total\text{-}DTT}$, we used three different methods of sample preparation using the HiVol quartz filters. Sample preparation schemes are illustrated in Fig. 1. Multiple method analysis was done only on HiVol filters since there was insufficient mass collected to compare different methodologies using the PCM Teflon samples.

**Method 1** consisted of two steps, water extraction and sequential methanol extraction. A 1-inch circular punch taken from the HiVol quartz filter was extracted in 4.9 mL DI via 30-minute sonication. The water extract was then filtered using a 0.45 $\mu$m PTFE syringe filter. This step was the same as the measurement of OP$^{WS\text{-}DTT}$. The water-extracted filter punch was retained in the vial, dried in room air and re-extracted using methanol (HPLC grade) via 30-minute sonication. The methanol





extract was also filtered through a syringe filter (0.45 $\mu$m PTFE) and then concentrated to about 200 $\mu$L using high purity nitrogen gently blown into the vial above the liquid surface. DI was added into the vial to reconstitute the small aliquot of remaining methanol liquid to 4.9 mL of solution. The reconstituted extract was stirred using a vortex mixer (VWR® Analog

Vortex Mixer, 300–3200 rpm) for 10 seconds to re-suspend any particles deposited on the walls of the vial during methanol blow-down. The purpose of the sequential and filtered methanol extraction was to assess if water-insoluble species could be dissolved by methanol as a way of quantifying the water-insoluble $OP^{DTT}$ through a contrast to methods that retained solid particles (discussed next). As methanol is less polar than water, it may dissolve most of the water-insoluble organic species in addition to some water-soluble compounds. However, since the solid-phase material in the extract may have been

removed by filtering the extract, this method will not include DTT-active species that cannot be separated from a solid particle and is therefore removed by the syringe filter. The determination of $OP^{DTT}$ for both water extract ($OP^{WS-DTT}$) and sequential DI-reconstituted methanol extract ($OP^{sM-DTT}$) was conducted using the $OP^{WS-DTT}$ analytical system since all extracts had been filtered, avoiding any plugging or contamination issues in the analytical system by solid particles. The sum of $OP^{WS-DTT}$ and $OP^{sM-DTT}$ is the total redox activity obtained by method 1, which will be denoted as $OP^{Total-DTT-1}$. Blank

filters were also similarly processed and analyzed for $OP^{DTT}$, producing blank values of $0.33 \pm 0.07$ nmol min$^{-1}$ (mean $\pm 1\sigma$, N=42) for $OP^{WS-DTT}$ and $0.43 \pm 0.09$ nmol min$^{-1}$ (mean $\pm 1\sigma$, N=18) for $OP^{sM-DTT}$. This method was used in the Southeastern Center for Air Pollution and Epidemiology (SCAPE) study and so a substantial data set (N=198) exists on $OP^{sM-DTT}$.

**Method 2** is similar to the methanol extraction by Yang et al. (2014). The filter punch was extracted in methanol via 30-minute sonication. After extraction, the filter punch was removed from the vial. The methanol extract was not filtered so that

the methanol-insoluble components were also retained and would possibly participate in the subsequent DTT reaction. The methanol suspension was blown down to nominally 200 $\mu$L using nitrogen gas and reconstituted to 4.9 mL with DI. The reconstituted extract was stirred for 10 seconds using a vortex mixer in an attempt to re-suspend particles deposited on vial walls. Due to the presence of solid material in the extract, such as quartz filter fibers released by sonication, the $OP^{WS-DTT}$ system could not be utilized. Instead, a modified automated system was needed to measure the OP of this aqueous

suspension, discussed below. The $OP^{DTT}$ of PM sample extracted in this manner is referred to as $OP^{Total-DTT-2}$. The blank value for this method was $0.42 \pm 0.13$ nmol min$^{-1}$ (mean $\pm 1\sigma$, N=18).

**Method 3** is the easiest to perform among the three methods in terms of sample preparation (Fig. 1). In this case the circular filter punch was immersed in the mixture of 4.9 mL DI and 1.4 mL K-buffer in a sterile polypropylene centrifuge tube, followed by 30min sonication. The DTT assay was then performed directly in the vial with the filter punch present using the

modified automated system discussed below. Some DTT-active species may be strongly absorbed to the filter surface so that they are not extractable into water. But in method 3, since the whole filter is suspended in DTT solution, these DTT-active species may participate in the reaction with DTT. In the study of Charrier et al. (2016), where DTT was also directly incubated with the PM filter, an alcohol, 2,2,2-Trifluoroethanol, was added to the extraction solvent to facilitate removal of



particles from the filter substrate. We tested adding small amounts of methanol (up to 10 % of total extraction volume) into
the extraction solvent to investigate if methanol would expose more solid aerosol for reaction with DTT, which would be
observed as an increase in DTT response. The test results are given in Supplement Fig. S4 and show that the added methanol
had negligible effects on the final $OP^{DTT}$ measured, therefore, only DI was used for extraction in this method. The $OP^{DTT}$
obtained in this way is referred to as $OP^{Total-DTT-3}$. Sonication versus shaking tests were also performed on method 3, and the
results (Supplement Fig. S5) show little effects of sonication on $OP^{Total-DTT-3}$ measurements. Only method 3 was used for the
$OP^{Total-DTT}$ determination of Teflon filters. Multiple blanks were processed similarly with DTT consumption rates of 0.37 ±
0.06 nmol min$^{-1}$ (mean ± 1$\sigma$, N=18) for quartz filters and 0.43 ± 0.04 nmol min$^{-1}$ (mean ± 1$\sigma$, N=18) for Teflon filters.

***Automated system for $OP^{Total-DTT}$ measurements:*** A modified automated analytical system for $OP^{Total-DTT}$ was developed by
modifying the $OP^{WS-DTT}$ system of Fang et al. (2015) for analysis of filters extracted using methods 2 and 3. A schematic is
shown in Fig. 2.  In this approach the sample extraction vial containing the suspension or suspension plus filter that had gone
through method 2 or 3 extraction is placed in the thermal mixer, prior to which 1.4 mL K-buffer had been loaded manually.
In this case, each sample vial is used as an incubation vial directly, continuously shaken and maintained at 37 ℃ via a
Thermo-mixer (VWR® Cooling Thermal Shake Touch; rotational frequency of 400 rpm, temperature of (37 ± 0.5) °C). Two
peek tubes (PEEK Tubing Green 1/16 inch OD × 0.030 inch ID), which are connected to a 14-port multi-position valve
(VICI Valco Instrument Co. Inc., USA), are inserted into each incubation vial, with one tube having an in-line syringe filter
(0.45 $\mu$m Polypropylene (PP) filter media, Whatman) and the other not. For each run, 0.7 mL DTT (1 mM) is loaded into the
incubation vial through the tubing without in-line filter via the programmable syringe pump A (see Fig. 2). Air is then
pumped into the incubation vial to thoroughly mix. In the mixture, DTT is presumably oxidized with the catalytic assistance
of both water-soluble and water-insoluble DTT-active species associated with the PM collected on the HiVol quartz or PCM
Teflon filter. After mixing, the multi-position valve is switched so that the syringe can withdraw an aliquot of sample
through the filter, at a low speed so as not to form air bubbles by cavitation. At designated time intervals (13, 30, 48, 65, 82
min), the aliquot is withdrawn through the in-line filter, transferred to the reaction vial (RV) and mixed with TCA preloaded
in the vial by pump B. The DTT concentration is then determined following the same steps as that for the $OP^{WS-DTT}$ system
(Fang et al., 2015). A total of five data points of remaining DTT concentrations versus time is generated and used for the
final $OP^{DTT}$ determination. After finishing the DTT analysis of each sample, the system is cleaned by flushing with DI to
remove the residual liquid left in the various tubing, reaction vial, pump syringes and LWCC. Following the flushing, the 14-
port multi-position valve is switched to the next sample for analysis. Due to the slow piston motions in liquid transfer from
IV to RV, it generally takes 1.5 hours for $OP^{Total-DTT}$ system to analyze one sample, compared with 1 hour of analysis time of
$OP^{WS-DTT}$. The $OP^{Total-DTT}$ system, like the $OP^{WS-DTT}$ system, can operate unattended and be monitored remotely to analyze, at
least, seven filters. (This is limited by the 14-channels of the multi-position valve in Fig. 2). To avoid contamination from the
insoluble material captured in the in-line syringe filter, the syringe filter is replaced after each sample run. The automated



system is cleaned every 4 weeks of continued operation by flushing at least three times with methanol, followed by four times with DI.

## 2.3 Other chemical analysis

A number of other aerosol components were analyzed on the HiVol quartz filters for assessing the various methods of measuring $OP^{Total-DTT}$. Carbon analysis (EC/OC) was performed on a 1.5 cm$^2$ punch from the quartz filters using Interagency Monitoring of Protected Visual Environments (IMPROVE) thermal optical reflectance (TOR) protocol (Chow et al., 1993).

Total and water-soluble metals were determined by inductively coupled plasma-mass spectrometry (ICP-MS) (Agilent 7500a series, Agilent Technologies, Inc., CA, USA) using EPA method 6020, again from sections of the same HiVol quartz filters. The elements of interest included K (potassium), Fe (iron), Mn (manganese), Cu (copper). For the determination of concentrations of total metals, acid digestion was carried out on a 1.5 cm$^2$ filter punch using nitro hydrochloric acid ($HNO_3$ + $3HCl$). The acid-digested sample was then diluted in DI water, filtered with a 0.45 $\mu$m PTFE syringe filter. No digestion was required prior to the analysis of water-soluble metals. A 1.5 cm$^2$ punch from the quartz filter was sonicated in DI for 30 minutes. After sonication, the extract was filtered using a 0.45 $\mu$m PTFE syringe filter, and then acid-preserved by adding concentrated nitric acid (70 %) to a final concentration of 2 % (v/v). A set of mixed calibration standard solutions were prepared by diluting the stock standard solutions and treated with the same procedures as samples. Internal standards including lithium ($^6Li$) and scandium ($^{45}Sc$), were added to all calibration standards and samples to monitor instrumental drift. DI blank and field blank which consist of same concentrations of acid and internal standards were used to monitor for possible contamination resulting from the sample preparation procedures. This was critical since in this case no special care was taken to pre-acid wash the quartz filters or syringe filters used in the water-soluble metals analysis. The method detection limits are defined here as three times the standard deviation of blanks, which for water-soluble metal method were 0.03 mg L$^{-1}$ for K, 0.00007 mg L$^{-1}$ for Mn, 0.009 mg L$^{-1}$ for Fe, 0.0002 mg L$^{-1}$ for Cu, and for the total metal method were 0.03 mg L$^{-1}$ for K, 0.0002 mg L$^{-1}$ for Mn, 0.02 mg L$^{-1}$ for Fe, and 0.002 mg L$^{-1}$ for Cu.

## 2.4 Data analysis

Linear regression was applied to the experimental data in order to assess relationships between measurements. Since the data was normally distributed (shown in Fig. S6), the Pearson correlation coefficients were calculated to further demonstrate the strength and the direction of a linear relationship between two measurements. A correlation coefficient greater than 0.7 with a low p-value ($<0.05$) was generally described as strong.

The paired t-tests were used to determine whether there was a significant difference in OP measurements between two methods. Each $OP^{Total-DTT}$ was measured using three methods, resulting in pairs of observations. The null hypothesis of the paired t-test assumed that the mean difference between the paired observations was zero. p-value of the test gave the



probability of observing the test results under the null hypothesis. p-values less than 0.05 rejected the null hypothesis at the 5 % significance level.

The F-tests in one-way analysis of Variance (ANOVA) were employed to evaluate the impact of filter type (i.e., quartz vs. Teflon filters) on the PM OP measurements for a given site. The F-statistic is the ratio of between-group variability to within-group variability, which followed an F-distribution under the null hypothesis. In this paper, the null hypothesis assumed that there was no significant OP difference between Teflon and quartz filters. If the F calculated from the data was smaller than the critical F-value of the F-distribution for significance level $\alpha=0.05$, then the null hypothesis would be true with 95 % confidence.

The spatial variability of OP (Table S5) was assessed by the coefficients of divergence (CODs) (Pinto et al., 2004; Wilson et al., 2005).

$$COD = \sqrt{\frac{1}{N}\sum_{i=1}^{N}\left(\frac{c_{ij}-c_{ik}}{c_{ij}+c_{ik}}\right)^2}, \qquad (1)$$

where $c_{ij}$ and $c_{ik}$ were $OP^{WS\text{-}DTT}$ or $OP^{Total\text{-}DTT}$ measured at site j and k, respectively, and N was the number of observations. A COD close to zero implied spatial uniformity, while a value approaching unity indicated absolute heterogeneity.

## 3. Results and discussion

First we discuss the performance of the automated system for measuring $OP^{Total\text{-}DTT}$ where filters were extracted by method 3 (Fig. 1), and then compare the results of the three differing methods for measuring $OP^{Total\text{-}DTT}$ at the two sampling sites. The system performance was assessed by only method 3 since these samples were easiest to prepare and this is the final approach of the three methods tested that was extensively utilized. Finally, we compare results from method 3 using quartz filters to a later study using Teflon filters. All $OP^{DTT}$ results were blank-corrected.

### 3.1 Automated $OP^{Total\text{-}DTT}$ system performance

The performance of the automated system was assessed in terms of the system response, accuracy and precision. 9,10-phenanthraquinone (PQN), a quinone that has been identified to be DTT-active (Kumagai et al., 2002) and often utilized as a positive control (Fang et al., 2015), was used to test the system response. A highly linear relationship ($R^2=0.97$) was found between PQN concentration in the incubation vial and the DTT consumption rate measured by the system (shown in Fig. 3). This linear relationship is consistent with the results shown in Fang et al. (2015) and Charrier et al. (2016).





The accuracy of measurements given by the OP$^{\text{Total-DTT}}$ system was further assessed by comparing the DTT consumption rate obtained by the system to that following the manual DTT analysis approach of Cho et al. (2005). Seven PQN solutions of various concentrations were tested by both the automated system and manual approach. As shown in Fig. 4, a bivariate linear regression was applied and yielded a slope near unity (0.99 ± 0.06), intercept close to zero (0.04 ± 0.04), and correlation of

determination (R$^2$) of 0.98. For further validation, five ambient samples, which in this case would include insoluble species, were extracted by method 3 and analyzed using both the automated and manual methods. The ratio of automated-to-manual DTT consumption rate was 0.98 ± 0.05. These tests illustrate the validity of the OP$^{\text{Total-DTT}}$ system as an alternative to the manual DTT assay.

To assess the precision of the automated OP$^{\text{Total-DTT}}$ system, the DTT consumption rates of identical concentrations of several

PQN solutions were repeatedly measured. The OP$^{\text{Total-DTT}}$ system produced consistent results for the PQN replicates (blank-corrected DTT consumption rate of 0.76 ± 0.05 nmol min$^{-1}$ for 0.21 nmol mL$^{-1}$ of PQN in the incubation vial, coefficient of variation (CV) = 6 %, N=7), suggesting good precision of the system. We conclude that most variability in the analysis of samples will be introduced in the extraction process and not the DTT analysis.

### 3.2 Precisions of various methods

To test the precision of the complete approach for measurement of OP$^{\text{Total-DTT}}$ (i.e., extraction and analysis), measurements of OP$^{\text{Total-DTT}}$ were repeated three times using three separate punches from the same Hivol quartz filter. This was done for all three OP$^{\text{Total-DTT}}$ methods. The coefficient of variation (CV) for replicates is used to assess the precision of each method. The results are summarized in Table 1. CV ranged from 3 % to 6 % for method 1, which may result from the combined uncertainties of the two respective steps (i.e., extraction and analysis). The range of CV for method 2 was from 5 % to 12 %.

The root of this variability may arise from the insoluble material remaining in the reaction suspension that was difficult to reproduce from run-to-run. In contrast, lower CV (1 %~5 %) was observed for method 3, possibly because it involved the least steps in the filter extraction.

### 3.3 Comparison of methods for measuring total oxidative potential (OP$^{\text{Total-DTT}}$)

### 3.3.1 Comparison of oxidative potential

In the following, OP$^{\text{DTT}}$ m$^{-3}$ determined by the three methods for simultaneously collected HiVol quartz filters at the GT and RS sites are compared. Since no standard method is available for assessing the ability to measure OP$^{\text{DTT}}$ m$^{-3}$, we simply compare the various methods and assume that the highest measurement represents the most comprehensive analytical method for measuring total oxidative potential. No HiVol conversion factors were applied to the OP data as the three methods were all performed on filters collected using the same HiVol sampler at each site.





Figure 5 shows the $OP^{DTT}$ m$^{-3}$ comparison between method 1 and 3 at both GT and RS sites. In general, the response of the DTT assay of method 3 was significantly higher than that of method 1 at the 95 % confidence level (paired t-test: p = 0.028 at GT, N=35; p<0.001 at RS, N=31). The results are expected since in method 1, both the water and methanol liquid extracts are filtered, potentially removing species that could have been DTT-active but remained attached to solid particles. The mean $OP^{Total\text{-}DTT\text{-}1}$ to $OP^{Total\text{-}DTT\text{-}3}$ ratio at GT was close to 1 (ratio = 0.95) and also higher than that at RS (ratio = 0.85). This may

imply that method 1 can be more effective for extracting aged PM species. The ratios of $OP^{sM\text{-}DTT}$ to $OP^{WS\text{-}DTT}$ are 0.34 ± 0.14 (N=35) at GT and 0.37 ± 0.12 (N=31) at RS, which are consistent with the ratios from SCAPE data (0.27 ± 0.08, N=198; unpublished data). The water-insoluble OP determined by the difference in $OP^{Total\text{-}DTT\text{-}3}$ (which includes solid particles) and $OP^{WS\text{-}DTT}$ ($OP^{WI\text{-}DTT\text{-}3}$ = $OP^{Total\text{-}DTT\text{-}3}$ - $OP^{WS\text{-}DTT}$) to $OP^{WS\text{-}DTT}$ ratio, by contrast, is 0.45 ± 0.25 at GT (N=35) and 0.67 ± 0.35 at RS (N=31). There was very little correlation between the $OP^{WI\text{-}DTT\text{-}3}$ and $OP^{WS\text{-}DTT}$ with Pearson correlations of

r = -0.23 and -0.51 at GT and RS sites, respectively (see Supplement Table S1). Similarly, $OP^{WI\text{-}DTT\text{-}3}$ had weak correlation with $OP^{sM\text{-}DTT}$ (Pearson correlation: r = 0.31 at GT; r = 0.04 at RS). Based on these data, it is clear that there were species associated with water-insoluble $OP^{DTT}$ not extracted by methanol and that remain attached to solid particles. This analysis shows that filtering the liquid extract, even if methanol solvent is used, will result in a substantial underestimation of $OP^{Total\text{-}DTT}$. Observations that $OP^{Total\text{-}DTT\text{-}3}$ is less than $OP^{Total\text{-}DTT\text{-}1}$ are likely due to propagation of errors for the summation method

(method 1) combined with variability in the extraction process for each method. Overall, these results indicate that method 3 is preferred to method 1 for measuring $OP^{Total\text{-}DTT}$.

Figure 6 shows the $OP^{DTT}$ m$^{-3}$ comparison between method 2 and 3 at both GT and RS sites. At the GT site, method 3 generally yielded higher OP responses compared to method 2 with mean $OP^{Total\text{-}DTT\text{-}2}$ to $OP^{Total\text{-}DTT\text{-}3}$ ratio of 0.90 (p<0.001 for a paired t-test (N=35)). For the RS site, however, method 2 was able to produce comparable (p=0.060 for a paired t-test,

N=31) or even higher OP responses, at times, than method 3 with a $OP^{Total\text{-}DTT\text{-}2}$ to $OP^{Total\text{-}DTT\text{-}3}$ ratio of 0.94, which may imply that method 2, in some cases, might be more efficient in extracting DTT-active species from the unique RS sources such as vehicular emissions. Overall, in terms of $OP^{DTT}$ responses, method 3 generally produced the highest signals compared to the other two methods, in both the urban (GT) and near-road (RS) sites.

### 3.3.2 Association between $OP^{DTT}$ and PM compositions

A correlation analysis was performed between measured PM$_{2.5}$ chemical constituents and $OP^{DTT}$ determined by the three methods. The results are summarized in Table 2. Compared to the $OP^{Total\text{-}DTT\text{-}1}$ and $OP^{Total\text{-}DTT\text{-}2}$, $OP^{Total\text{-}DTT\text{-}3}$ is moderately to strongly correlated with many of the measured species (r=0.48–0.82 for GT data, r=0.49–0.72 for RS data). In contrast, $OP^{Total\text{-}DTT\text{-}2}$ is correlated with the least number of measured PM species. $OP^{Total\text{-}DTT\text{-}1}$ and $OP^{Total\text{-}DTT\text{-}3}$ are correlated with more species than $OP^{WS\text{-}DTT}$, suggesting that they capture more chemical components contributing to DTT than $OP^{WS\text{-}DTT}$. By

subtracting $OP^{WS\text{-}DTT}$ from $OP^{Total\text{-}DTT}$, $OP^{WI\text{-}DTT}$ is determined for the three methods. Among the methods, only the water-insoluble $OP^{DTT}$ determined by method 1, i.e. $OP^{sM\text{-}DTT}$, has good correlation with PM species. $OP^{sM\text{-}DTT}$ is strongly





correlated with OC, EC and Cu (both water-soluble and total) at GT and OC, EC and water-soluble Fe at RS. Verma et al. (2012) also showed good correlation between $OP^{DTT}$ of filtered methanol extracts and OC and EC, and attributed this association to water-insoluble organic carbon species (WIOC) that dissolve in methanol. Thus, $OP^{sM-DTT}$ in method 1 is

likely attributed to some fraction of the WIOC. $OP^{sM-DTT}$ is a direct measure of $OP^{WI-DTT}$, whereas $OP^{WI-DTT}$ is determined by difference for method 2 and 3, which leads to larger uncertainty and more scatter associated with these data. In a subsequent study, discussed next, only Method 3 was utilized to measure $OP^{Total-DTT}$ of PM for Teflon filters because of its better precision, more comprehensive response, association with greater numbers of PM components, and easiest filter preparation (extraction) process.

**3.4 $OP^{WS-DTT}$ and $OP^{Total-DTT}$ measurements on quartz versus Teflon filters and their spatial distributions**

The time series of volume-normalized water-soluble and total $OP^{DTT}$ via method 3 are shown in Fig. 7 for two different sample time periods using HiVol samplers with quartz filters (21 April 2016–30 May 2016; HiVol conversion factors of 1.00 and 1.10 were applied to GT $OP^{WS-DTT}$ and $OP^{Total-DTT}$ data, respectively.) and PCMs with Teflon filters (26 July 2016–21 August 2016). A summary of the average OP data is given in Supplement Table S2. The ANOVA results (Supplement Table

S3) indicate negligible difference between types of filter (i.e., quartz versus Teflon) on $OP^{DTT}$ measurements.

Figure 7 shows that, as expected, $OP^{Total-DTT}$ is always higher than $OP^{WS-DTT}$. The ratios of $OP^{WS-DTT}$ to $OP^{Total-DTT}$ were on average $65 \pm 10$ % and $65 \pm 14$ % at GT compared to $62 \pm 12$ % and $58 \pm 10$ % at RS, for quartz and Teflon PM samples, respectively. Thus, $OP^{Total-DTT}$ of $PM_{2.5}$ contained on average 35 to 42 % insoluble species. The correlation coefficients between $OP^{WS-DTT}$ and $OP^{Total-DTT}$ were 0.71 and 0.56 for quartz filters at GT and RS, respectively (Table S1). These

moderate correlations reflect some contribution of insoluble species to total OP.

Spatial distributions in $OP^{DTT}$ can also be investigated. As discussed above, the water-soluble fraction of total OP ($OP^{WS-DTT}$ to $OP^{Total-DTT}$ ratio) was fairly similar at the two sites, which means that the insoluble fraction was not vastly different between the two sites. Figure 8 shows a summary of daily concentration ratios between the sites. EC, a marker for incomplete combustion, and so associated with vehicle emissions, was much higher at the RS site; the ratio of RS to GT was

3.2 on average. OC was only slightly elevated, as expected, since OC is largely secondary in Atlanta (Xu et al., 2015) and so more spatially uniform (i.e., primary OC is a small fraction of total OC, even at RS). Both $OP^{WS-DTT}$ and $OP^{Total-DTT}$ were spatially uniform with daily RS-to-GT OP ratios close to one. CODs were also calculated to further assess the spatial variability of OP (Table S5). The low COD values (COD<0.08 for the quartz filters and <0.23 for the Teflon filters) between RS and GT site indicate spatial homogeneity of OP during the sampling periods. This was found for both quartz and Teflon

filters. The homogenous distributions of OP are very similar to that of OC (COD=0.18) and in contrast to EC (COD=0.52). Note that both $OP^{WS-DTT}$ and $OP^{Total-DTT}$ were slightly higher at the RS site, possibly indicating a linkage to RS emissions. Uniformity of $OP^{WS-DTT}$ is consistent with the results shown in the study of Fang et al. (2015), but similar uniformity in



$OP^{Total-DTT}$ may seem somewhat unexpected since water-insoluble aerosol components are often associated with primary species. These data show that even $OP^{WI-DTT}$ species are largely secondary. These results are consistent with studies that have found water-insoluble DTT-active constituents could be secondary quinones from oxidized PAHs that remain bound to the surface of soot particles associated with traffic emissions (Antinolo et al., 2015; Li et al., 2013; Shiraiwa et al., 2012). This means that although roadway emissions are a source for components that contribute to OP, some form of processing is needed to convert the roadway emissions to species with measurable oxidative potential for both $OP^{Total-DTT}$ and $OP^{WS-DTT}$. Size distributions of $OP^{WI-DTT}$ (Fang et al., in preparation) suggest that $OP^{WI-DTT}$ is composed of different types of insoluble species, OP from oxidized aromatic species (e.g., quinones) may be mainly associated with smaller sized insoluble soot particles, and at the large end of the $PM_{2.5}$ size range, transition metal ions (i.e., water-soluble Cu) associated with road and brake dust may be the main source.

## 4. Summary

An automated analytical system was developed for quantifying total aerosol oxidative potential with the DTT assay ($OP^{Total-DTT}$) from filter sample extracts. The method is based on modifying an automated analytical system developed by Fang et al (2015) for measuring water-soluble oxidative potential ($OP^{WS-DTT}$). Three methods for including the contribution of water-insoluble components to oxidative potential of PM ($OP^{WI-DTT}$), for a measurement of $OP^{Total-DTT}$ were tested: 1) Extracting filter punches in deionized water (DI), filtering the extract and measuring $OP^{WS-DTT}$, followed by methanol extraction on the same filter, filtering the extract and removing most methanol by evaporation, then reconstituting in water and summing with $OP^{WS-DTT}$ to obtain $OP^{Total-DTT}$; 2) Extracting filter punches in methanol, reconstituting the unfiltered methanol extracts with DI after evaporation of methanol, and performing the DTT assay on the DI-reconstituted suspension; 3) Extracting filter punches in a vial with DI and then performing the DTT assay in the vial containing the filter. Method 3 generally yielded higher DTT responses with higher precision (coefficient of variation of 1~5 %), and was highly correlated with more aerosol species, including OC, EC and various water-soluble and total elements. Because this method requires no use of organic solvents that must be mostly eliminated prior to DTT analysis, it is the easiest to automate. The automated system for measuring $OP^{WS-DTT}$ (Fang et al., 2015) was modified to follow method 3 and the system performance was tested.

An ambient study was conducted to contrast measures of $OP^{Total-DTT}$ and $OP^{WS-DTT}$ for $PM_{2.5}$ collected at a roadside (RS) site (highway with restricted heavy duty diesel access) and a site more representative of overall average urban Atlanta air quality (GT). Simultaneous daily filter samples were collected during two separate one-month periods and comparisons were made using quartz and Teflon filters. At the representative urban site (GT), the ratio of $OP^{WS-DTT}$ to $OP^{Total-DTT}$ was 65 % for both types of filters. At the roadside site (RS) the ratio was only slightly lower, 62 % for quartz filters, 58 % for Teflon filters. $OP^{WS-DTT}$ and $OP^{Total-DTT}$ were moderately correlated with Pearson Product correlation coefficients between 0.56 (roadside) and 0.71 (urban). Simultaneous measures of $OP^{WS-DTT}$ and $OP^{Total-DTT}$ at the GT and RS site showed only slightly higher levels of both at the RS site, indicating both $OP^{WS-DTT}$ and $OP^{Total-DTT}$ were largely spatially homogeneous. The results are



consistent with roadway emissions as sources of OP, but that $PM_{2.5}$ OP was largely secondary for both soluble and insoluble aerosol components contributing to OP.

**Acknowledgements**

We would like to thank Emily Saad for ICP-MS analysis. This work was funded through a Research Agreement with the Health Effects Institute (#4942-RFA13-1/14-3) and EPA STAR Grant R834799 that supported the Southeastern Center for
Air Pollution & Epidemiology (SCAPE). The contents are solely the responsibility of the grantee and do not necessarily represent the official views of the sponsors. Further, US EPA does not endorse the purchase of any commercial products or services mentioned in the publication.

**Competing interests**

The authors declare that they have no conflict of interest.

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



**Table 1.** Coefficient of variation (CV) of $OP^{Total-DTT}$ for three extraction methods.

|  | Method 1 | Method 2 | Method 3 |
|---|---|---|---|
| Coefficient of variation (CV) | 3–6 % N=10 | 5–12 % N=7 | 1–5 % N=12 |

*N is the number of HiVol filters tested.


**Table 2**. Pearson's r between $OP^{DTT}$ $m^{-3}$ and PM chemical components at GT (N=34) and RS (N=29) sites.

| GT |  | OC | EC | WS | | | | Total | | | |
|---|---|---|---|---|---|---|---|---|---|---|---|
|  |  |  |  | K | Mn | Fe | Cu | K | Mn | Fe | Cu |
| Method1 | $OP^{WS-DTT}$ | **0.79\*\*** | **0.84\*\*** | 0.63\*\* | 0.46\* | 0.49\* | **0.77\*\*** | 0.53\*\* | 0.43\* | 0.36 | **0.78\*\*** |
|  | $OP^{sM-DTT}$ | **0.71\*\*** | 0.66\*\* | 0.28\* | 0.44\*\* | 0.45\*\* | 0.4\* | 0.35\* | 0.38 | 0.2 | 0.37\* |
|  | $OP^{Total-DTT-1}$ | **0.76\*\*** | **0.81\*\*** | 0.51\*\* | 0.46\* | 0.54\*\* | **0.74\*\*** | 0.39\* | 0.41\* | 0.38\* | **0.72\*\*** |
| Method2 | $OP^{Total-DTT-2}$ | 0.51\*\* | 0.44\* | 0.27 | 0.40\* | 0.09 | 0.62\*\* | 0.15 | 0.37\* | 0.23 | 0.59\*\* |
|  | $OP^{WI-DTT-2}$ | 0.44\* | 0.25 | -0.31 | 0.46\* | -0.39 | 0.53\*\* | -0.28 | 0.32 | 0.13 | 0.53\*\* |
| Method3 | $OP^{Total-DTT-3}$ | 0.66\*\* | **0.78\*\*** | **0.82\*\*** | 0.69\*\* | 0.48\* | **0.76\*\*** | 0.69\*\* | **0.73\*\*** | **0.71\*\*** | **0.78\*\*** |
|  | $OP^{WI-DTT-3}$ | 0.44\* | 0.48\*\* | 0.55\*\* | 0.57\*\* | 0.26 | 0.38 | 0.50\* | 0.66\*\* | 0.63\*\* | 0.56\*\* |
| RS |  |  |  |  |  |  |  |  |  |  |  |
| Method1 | $OP^{WS-DTT}$ | **0.83\*\*** | **0.79\*\*** | 0.67\*\* | 0.43\* | **0.88\*\*** | 0.54\*\* | 0.69\*\* | 0.48\* | 0.57\*\* | 0.40 |
|  | $OP^{sM\_DTT}$ | **0.72\*\*** | **0.72\*\*** | 0.48\*\* | 0.13 | **0.73\*\*** | 0.65\*\* | 0.59\*\* | 0.18 | 0.53\*\* | 0.38 |
|  | $OP^{Total-DTT-1}$ | **0.77\*\*** | **0.76\*\*** | 0.67\*\* | 0.28 | **0.80\*\*** | 0.63\*\* | 0.67\*\* | 0.39\* | 0.61\*\* | 0.42\* |
| Method2 | $OP^{Total-DTT-2}$ | 0.68\*\* | 0.52\*\* | 0.53\*\* | 0.45\* | 0.09 | 0.44\* | 0.48\* | 0.59\*\* | 0.57\*\* | 0.65\*\* |
|  | $OP^{WI-DTT-2}$ | 0.68\*\* | 0.51\*\* | 0.50\* | 0.48\* | 0.07 | 0.42\* | 0.42\* | 0.61\*\* | 0.66\*\* | 0.66\*\* |
| Method3 | $OP^{Total-DTT-3}$ | **0.71\*\*** | 0.68\*\* | 0.6\*\* | 0.56\*\* | 0.49\* | 0.55\*\* | 0.67\*\* | 0.66\*\* | 0.66\*\* | **0.72\*\*** |
|  | $OP^{WI-DTT-3}$ | -0.34 | -0.37 | -0.47\* | -0.37 | -0.40\* | -0.40\* | -0.43\* | 0.31 | -0.39\* | 0.43\* |

Note: Water-insoluble $OP^{WI-DTT}$ obtained by method 2 and 3 is calculated by $OP^{WI-DTT} = OP^{Total-DTT} - OP^{WS-DTT}$.
r>0.70 are bold.
\*\*p-value<0.01. \*p-value<0.05. The correlation not statistically significant is without superscript and greyed out.




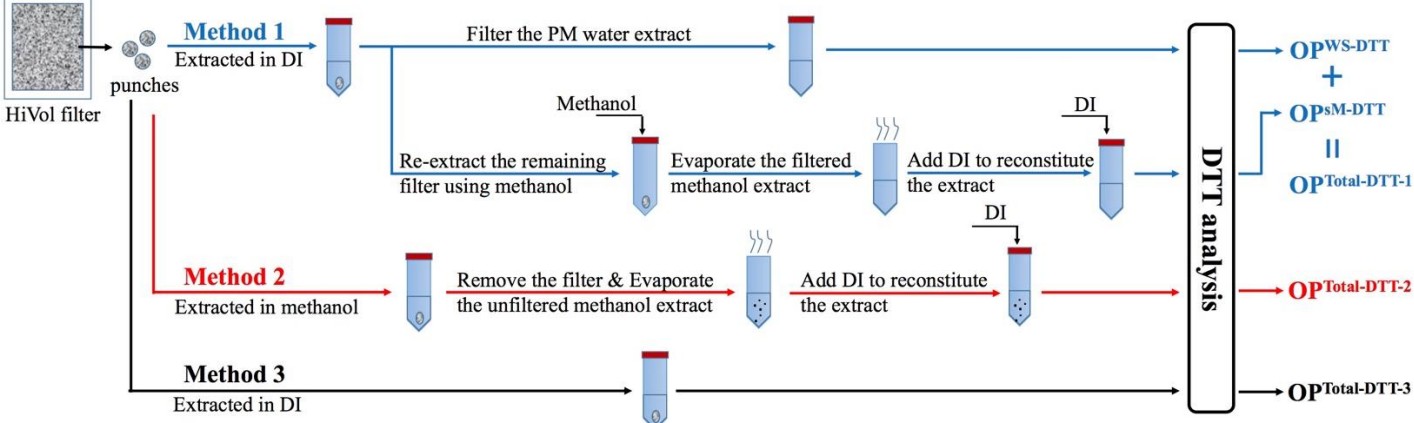

**Figure 1.** Analytical scheme for three sample extraction methods to determine total OP with the DTT assay (OP$^{Total\text{-}DTT}$).


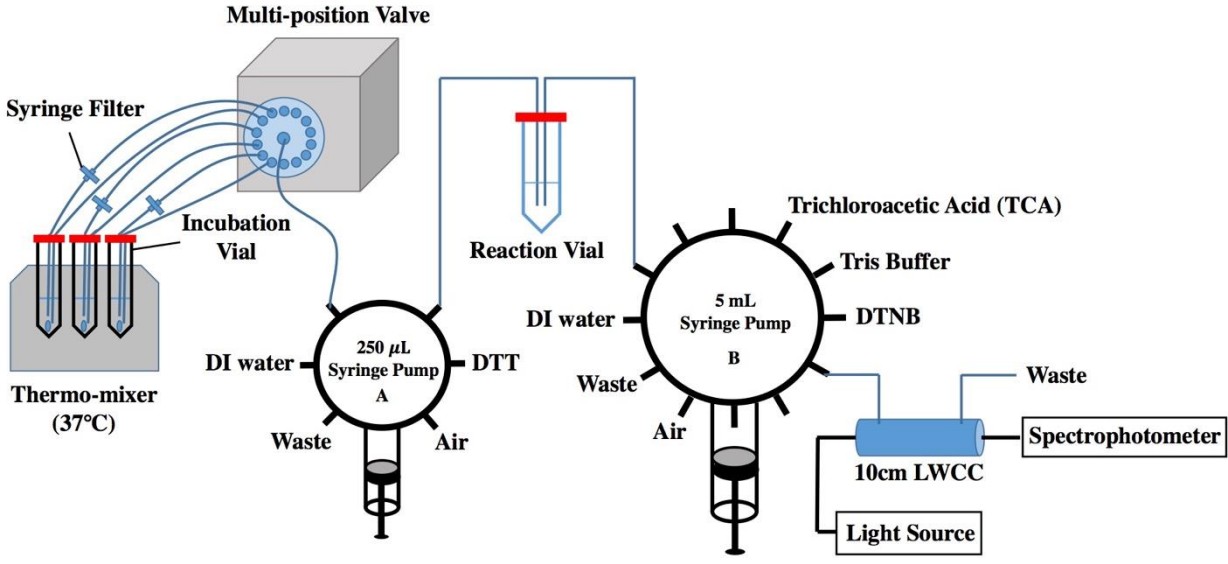

**Figure 2.** Automated system setup for measuring OP$^{DTT\text{-}Total}$. The assay is performed in the vial containing the filter sample and extraction water, which had been sonicated. The assay is filtered just prior to analysis in the liquid wave guide capillary cell (LWCC).





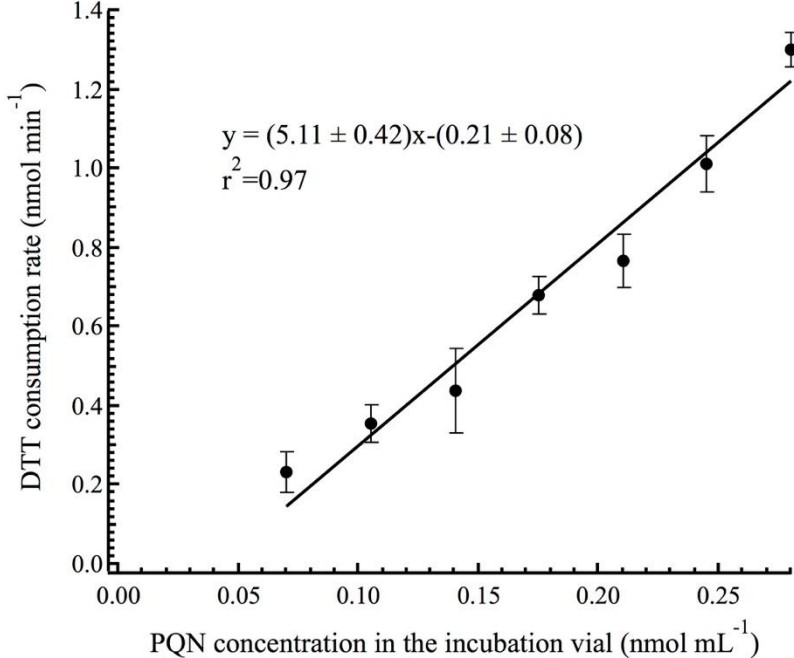

**Figure 3**. Blank-corrected DTT consumption rate as a function of PQN showing linearity between PQN concentrations and DTT consumption rate for the total analytical system (for PQN levels shown in the range above). Error bar represents the standard deviation of three independent DTT measurements on each concentration.





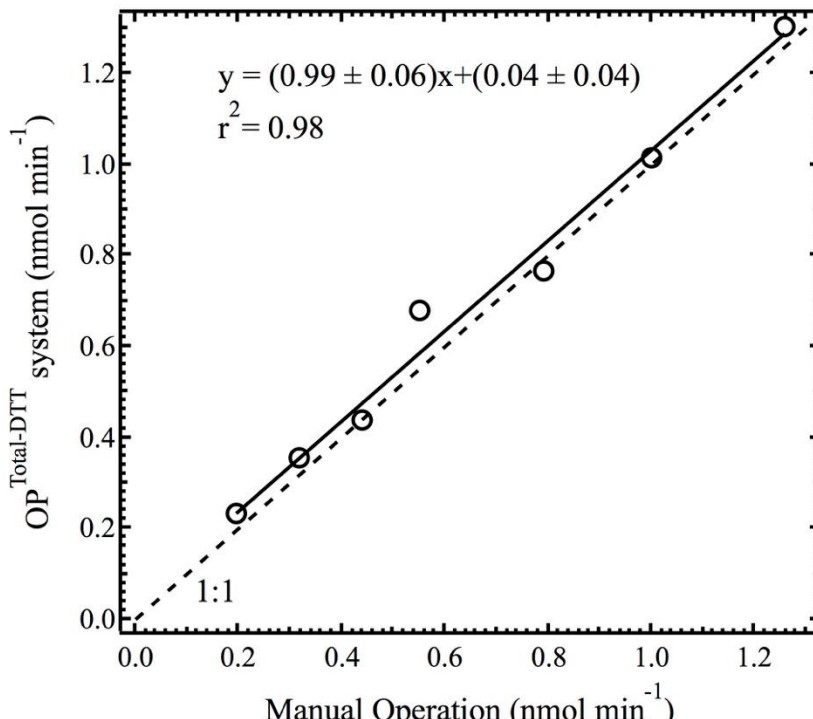

**Figure 4**. DTT consumption rate (blank-corrected) comparison of the automated system for measuring OP[Total-DTT] (shown in Fig. 2) to a
manual analysis using PQN (9,10-phenanthraquinone). Slope (± 1 standard deviation) and intercept (± 1 standard deviation) are based on
orthogonal regression.





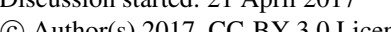

**Figure 5.** Comparison of $OP^{DTT}$ m$^{-3}$ between extraction method 1 and 3 at (a) GT (N=35) and (b) RS (N=31). Error bars denote one standard deviation in $OP^{DTT}$ m$^{-3}$ from repeated measurements and are propagated in calculating $OP^{Total-DTT-1}$.





**Figure 6.** Comparison of $OP^{DTT}$ m$^{-3}$ between method 2 and 3 at (a) GT (N=35) and (b) RS (N=31). Error bars denote one standard
deviation in $OP^{DTT}$ m$^{-3}$ from repeated measurements.



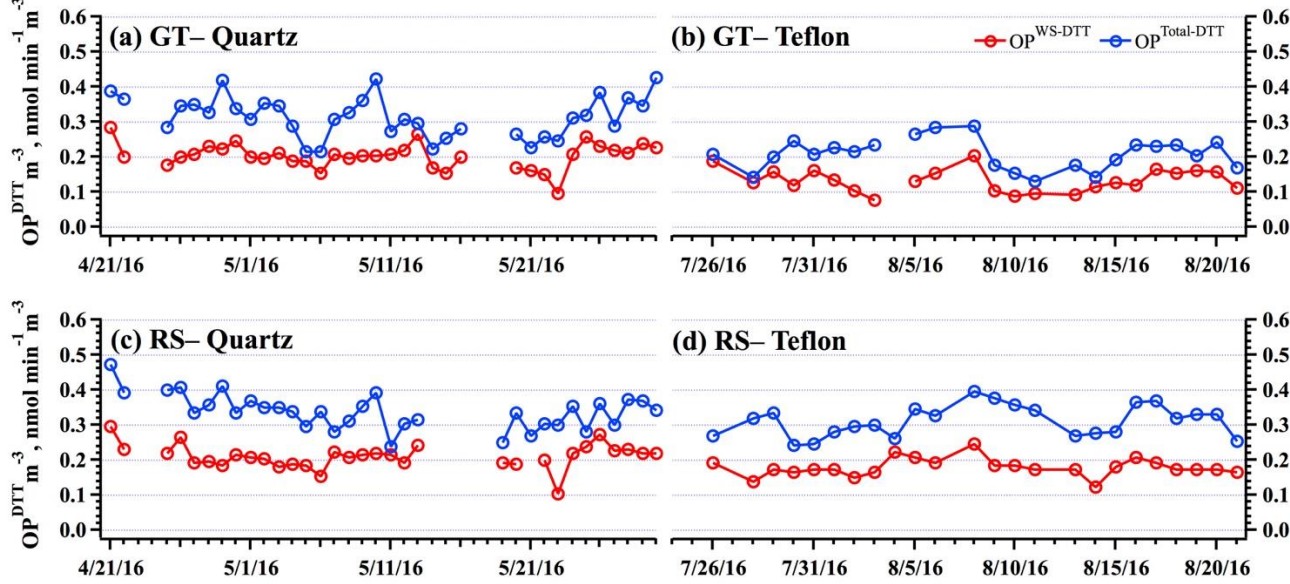

**Figure 7.** Volume normalized $OP^{DTT}$ of ambient $PM_{2.5}$ particles collected on quartz and Teflon filters at GT and RS sites for two different sampling time periods. Red lines indicate volume normalized $OP^{WS-DTT}$, and blue lines denote volume normalized $OP^{Total-DTT}$.





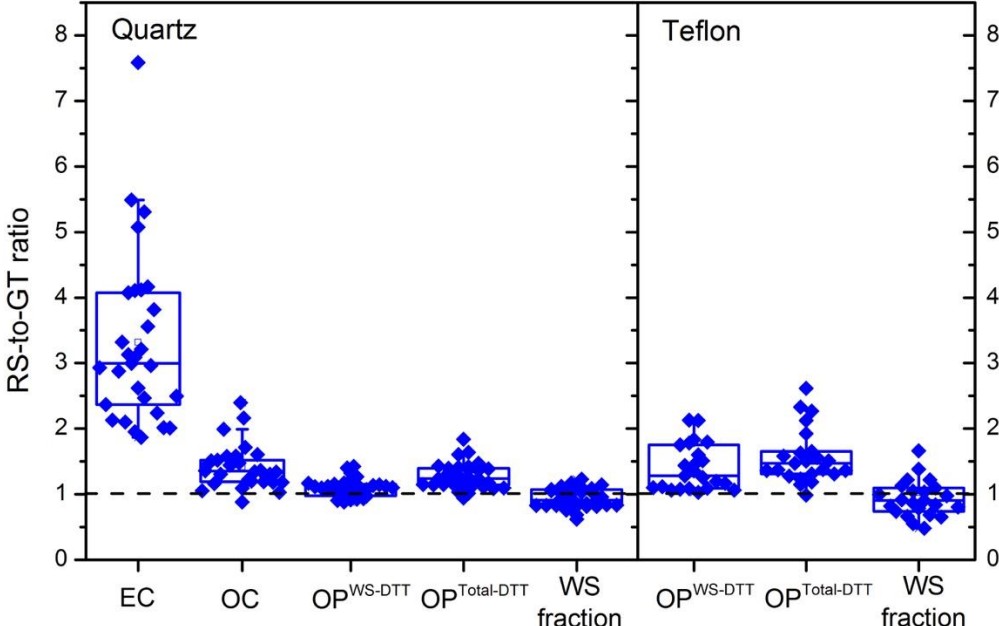

**Figure 8.** Comparison of simultaneous measurements at GT and RS sites based on daily RS-to-GT concentration ratios. The bottom and top of the box are the first (Q1) and third quartiles (Q3), and the band inside the box is the median. The lowest and highest ends of whisker are (Q1-1.5IQR) and (Q3+1.5IQR), where the interquartile range IQR=Q3-Q1.