# Peer review of "A method for measuring total aerosol oxidative potential (OP) with the dithiothreitol (DTT) assay and comparisons between an urban and roadside site of water-soluble and total OP"

_Atmospheric Measurement Techniques, 2017_

## Referee Comment (RC1) · Anonymous Referee #1 · 8 May 2017

General comments: It is important to characterize and identify the components of PM contributing to its oxidation potential. Time-resolved and automated analysis systems are important to better assess exposure effects. The paper addresses the need for faster laboratory results and compares different methodologies to identify the relevant fractions contributing to PM oxidative potential. However, the methodology applied in some cases is not well thought and important points have not been addressed in the manuscript or considered by the authors. Thus, some of the conclusions are not correct, unless several assumptions are made.

[Figure]

Main Comments: Line 53: authors note that water soluble OP is the common focus since it is the most straightforward. This reviewer disagrees as total DTT is most commonly reported and is a direct measurement, easier to conduct than water soluble OP, as no extraction, filtration or phase separation is required to conduct the analysis. Line 100: System comparison is important. Identifying the bias between collection systems is vital to identify and correct for differences in the results. However, as both systems are collecting PM in similar conditions, how can authors explain that there is no difference for WS-DTT but they have a considerable 10% for total DTT? The authors should indicate a possible explanation for this difference, is this due to the extraction? It is difficult to imagine a sampling difference that will only affect the non-water soluble components. Line 116. Authors indicate that the results from the comparison between extraction by sonication or shaking are similar; although the correlation is reasonable, the scattering is significant. This variation with extraction may indicate an effect of the extraction method depending on the chemical composition of the PM collected and extracted. Line 150. Method 1. After extraction with water and resuspension of the extracted solution, there is a considerable amount of sample removed from the filter. These particles are not then extracted with methanol which may lead to underestimating the OP of the methanol extract. Did the authors estimate the loss of particle mass during the first extraction? How much of the original sample remain in the vial for the consecutive extraction? It will be important to know how much is lost if corrections are to be made for more accurate comparison of the OP of the different fractions. Thus, the OP-Total DTT1 (line 164) obtained by adding the OP of each fraction is not accurate and already biased for comparison with the other extraction methods. To obtain a more accurate measurement of the OP of each fraction, and as result the total obtained using this method, the vial containing the extracted sample could be centrifuged and the supernatant removed. This way resuspended particles will remain in the vial for further extraction. Particles can be resuspended by short vortex mixing, and the sample ready for methanol extraction. Line 177. Method 2. Again, there is a bias in the method as a considerable fraction of the collected PM is removed with the filter.

Depending on the particle size, extraction efficiency varies. What was the extraction efficiency in this case? How much of the sample was removed with the filter? Ultrafine particles are not extracted from the filters easily and as many studies have shown, they are usually the main contributors to the OP of PM. As a result, the measurement is not accurate and comparison with other methods is not valid. Corrections could be made if the efficiency was known and OP measurements adjusted accordingly. Line 192. Automated system. In the description of the analytical method authors indicate that the system lines, valves and pump are only clean after a sample is run, have the authors considered interference in the results by cross contamination and sample interference between consecutive injections of the same sample? The solution remaining in the lines, ports and valves could interfere with the signal of the next injection. As the time between injections is considerable, reaction between DTT and solution can continue while in the lines and valves. Line 305. This sentence is speculation. Unless a comprehensive chemical analysis is conducted to identify the nature of the compounds associated with the sample, the statement is authors speculation. Line 311. Based on pervious comments made regarding the methods, this statement may not be correct. There is considerable amount of sample removed from the vial prior to the methanol extraction, which may lead to underestimation of the real potential of the insoluble components. If authors did not quantify the particle loses the methanol extract OP cannot be assumed accurate. Line 335. The OPsM-DTT is not a direct measure of the oxidation potential of water insoluble component. Components associated with PM present different solubility in different solvents. Many components are not extracted by methanol, and required other organic solvent for extraction. Extraction with hexane, dichloromethane, or acetonitrile, as examples, may result in different OP of the insoluble fraction. Thus, this reviewer disagrees with the statement made by the authors. It will be very difficult to directly measure the OP associated with the insoluble material. The more accurate measurement will be an estimation based on the difference between total OP and water-soluble (filtered) OP. Line 350. A contribution of 35 to 42% of the insoluble material to the total OP is not "some contribution", it is a significant

fraction of the total OP. Line 364. This sentence is again speculation. The authors do not present any results regarding the chemical composition to support the idea of secondary compounds as main contributors to the oxidative potential. Identification of primary and secondary compounds, as well as correlation between the different fractions, is required to validate this statement. Wind pattern, local emission sources and measurements at each location are needed to indicate the presence and contribution of primary and secondary compounds. Line 394. OP measurements for two points, GT and RS, are not representative of a wide area, so concluding the measured OP were largely spatially homogeneous is too broad of a statement. Line 395. Again, this is speculation by authors. Unless the correlation between primary and secondary compounds with OP is presented, there are no bases to make this statement.

Line 520. Table 2. Authors present correlation between OP measured under different methods and chemical speciation. Among the chemical compounds presented by the authors, K shows important correlation with OP. However, this compound has not been shown to be redox active or a chemical that can react with DTT. Why do the authors include this compound in their list and their correlations? If this is not a redox compound it does not contribute to the OP, and the correlation is not significant; unless the compound is used as a marker for sources that can contribute to other compounds that do contribute to the OP of the sample. A coherent explanation is needed regarding the inclusion of K in the table and results.

---

## Referee Comment (RC3) · Anonymous Referee #4 · 10 May 2017

Review for A method for measuring total aerosol oxidative potential (OP) with the dithiothreitol (DTT) assay and comparisons between an urban and roadside site of water-soluble and total OP by Dong Gao et al.

This manuscript describes a careful study further developing a technique previously published by the group. Different aerosol sampling and extraction methods are compared and first field data are presented to quantify aerosol bound OP. After addressing the points listed below, I recommend publication in AMT.

p.2, line 37: OP(total-DTT) is that the unfiltered methanol extract fraction? It might be

[Figure]

best not to use undefined abbreviations in the abstract.

p. 2, line 40-41: This sentence is not clear to me. Was the same DTT analysis performed on Teflon filters? Please be more specific.

p. 3, line 62/63: It would be good to mention also a reference for the DCFH assay besides DTT and ascorbic acid.

p.8, line 167 – 169. I do not agree with the statement that sonication has no effect on radical formation. The paper cited, Miljevic demonstrates explicitly the opposite. Fig. 3 in that paper shows clearly that sonication strongly oxidises DTT! It is not clear to what section in the SI this sentence refers to. Fig. S3 or S5? In Fig S5 both axes are labelled that same. So it is not clear what Fig S5 is showing. This statement has to be worded much more carefully and the results presented and the potential effects of sonication have to discussed critically by representing literature results correctly!

p. 10, line 223: I would strongly advice to call the "water-insoluble OP" "methanol-exracted OP" as this is highly misunderstanding. There are many organic compounds that are not water soluble but are also not soluble in methanol, such as many PAHs. By reconstituting the methanol extract again in water, it is likely that components which are not water soluble precipitate again and are thus not accessible to the DTT oxidation anymore. This potential artefact should be discussed.

p. 11, line 241-247: Filtration of quartz fibre filters usually results in significant disintegration of the filter and many loose quartz fibres in the extract. As these samples were not filtered, it should be described in more detail how this was dealt with and what potential artefacts this might have caused for the DTT analysis.

p. 11, line 252: Why was K-buffer used in method 3 but not in the other methods? Please explain this differences and potential consequences for the results.

p. 18, line 398: Please check all "WS-DTT" and "WI-DTT" super-scripts. It seems to me there are some typos.

p. 19, Table 2: Was there any difference observed in the correlation between DTT values and total and water soluble metal concentrations?

---

## Referee Comment (RC4) · Anonymous Referee #3 · 15 May 2017

This article reported three methods of analyzing aerosol oxidative potential (OP) based on DTT assay, focusing on different sample extraction strategies to distinguish soluble components from the total aerosol particulate matter (PM). I think this is a good work with practical meaning to assist the understanding of oxidative aerosol's transport in atmosphere. I have one major suggestion as below.

Figure 1 delivered a clear demonstration of the three sample extraction and analysis methods. However, I think a better job can be done to explain what component(s) from the total PM species is measured by each method throughout the text. Basically, two factors, including extraction solvent (water versus methanol) and filter (i.e., with or without filtering) differentiate the three methods. Which component in the aerosol is screened by each factor should be clearly specified to elucidate what is the actual difference across the three methods. From my understanding, the use of water or methanol discriminates the water-soluble (herein "soluble" matter includes dissolved molecules/ions and "dispersed" solids (or small particles)) and methanol-soluble component, whereas filtering or not filtering a sample discriminates dissolved molecules/ions and dispersed solids in that sample. Overall, extraction combined with these two factors actually categorizes the total PM into six groups:  water-insoluble species (those which cannot be extracted into water), water-soluble molecules/ions, water-soluble solids (dispersible small particles), methanol-insoluble species (those which cannot be extracted into methanol), methanol-soluble molecules/ions, and methanol-soluble solids. The authors should clarify which groups are measured in each of the three methods, either with a diagram or plain text.

Elucidating the above may rationalize a few ambiguous places better in the articles. What follows are some examples.

L59-60,76-77, there seem to be some conflicts between these two references: the first place says up to 99% of the DEP CANNOT be extracted by water or methanol while the second one says the measurement is based on water extraction. "after which the filter was removed from the analysis" is very confusing. What are the authors trying to say here?

L100-101: what is "10%" here?

L300-323: The justification of choosing method 3 over the other two is poor. It maybe improved if the authors can specify each species category corresponding to each separation technique as I suggested.  I have specific questions as below:
         L305-309, it seems method 1 is more consistent with SCAPE study, than method 3; then why is method 3 selected over method 1?
         L309, what is indicated by "very little correlation between $OP^{WI-DTT-3}$ and $OP^{WS-DTT}$?
         L315, I am uncomfortable with "overall" here, because the only justification of method 3 over method 1 is that it measures higher total-DTT (which is actually

quite common sense). The whole paragraph seems quite over-informative and little relevant to the conclusion.

L319-323, the same problem is with the comparison of method 2 and method 3. There is not actual justification for why method 3 is chosen over method 2. Information is very redundant and little relevant to the conclusion.

L330-339, it's somewhat ironic here. I was told that method 1 is better than method 2 and 3 in terms of seeking correlation between OP-DTT and PM compositions but then suddenly method 3 was selected to be used... Why?

L348: I recommend clarifying how 35-42% is determined and specify the error range as well.

Overall, I strongly suggest the authors give their recommendations on which method should be used in what scenarios or for what purposes.

Other minor suggestions:

L133-134: provide possible explanations on what gives the response in blank samples.

L275-278: Maybe provide a figure in SI to illustrate the validation with five ambient samples as well?

Table 1: Does "N" filters correspond to different samples? (I assume CV is determined with three replicates on each filter (each sample)?) and what is the range standing for.

Table 2: I think a figure is better to let readers see the correlations of different variables, although I know there are many comparisons here. Maybe the authors can think if the illustration here can be improved.

---

## Author Comment (AC1) · 7 Jul 2017

*We thank the reviewers for their comments. Our responses and corresponding changes made in the manuscript (highlighted in red) are given below.*

**Response to anonymous referee #1 comments:**

**General comments: It is important to characterize and identify the components of PM contributing to its oxidation potential. Time-resolved and automated analysis systems are important to better assess exposure effects. The paper addresses the need for faster laboratory results and compares different methodologies to identify the relevant fractions contributing to PM oxidative potential. However, the methodology applied in some cases is not well thought and important points have not been addressed in the manuscript or considered by the authors. Thus, some of the conclusions are not correct, unless several assumptions are made.**

**Main Comments:**

1) **Line 53: authors note that water soluble OP is the common focus since it is the most straightforward. This reviewer disagrees as total DTT is most commonly reported and is a direct measurement, easier to conduct than water soluble OP, as no extraction, filtration or phase separation is required to conduct the analysis.**

    **Response:** *We believe it is a large assumption that one can comprehensively measure all water-insoluble components in a detection system that is based on transferring particles collected on a solid surface (filter) that are then transferred to a liquid for performing the chemical assay and the subsequent quantitative analysis. An a priori assumption that this can be done with 100% efficiency is likely a poor assumption. We believe it is much easier (and more likely to be closer to 100% efficient) to extract a filter with water, filter the sample and then perform the analysis in water to comprehensively measure just the water-soluble species. As proof we note that one does not find any systems (that we know of) for measuring total carbonaceous aerosol concentrations based on liquid systems, but measurement of water-soluble organic carbon is a widely accepted practice. (See issues with particle adhesion, brought up by this reviewer, below). As noted in the manuscript, there is no consistent approach to determine total $OP^{DTT}$, and some methods are highly complex involving the evaporation of organic solvents. There have also been no reported intercomparisons of the various methods used and no widely accepted standard method for measuring total OP, making it difficult to compare the results between studies. These are the exact issues addressed in this paper. The challenge is that there is no gold standard for comparison, so we are left with comparisons between methods. For the sake of not making too strong a statement, we have deleted "since it is the most* straightforward to measure*" in line 53.*

2) **Line 100: System comparison is important. Identifying the bias between collection systems is vital to identify and correct for differences in the results. However, as both systems are collecting PM in similar conditions, how can authors explain that there is no difference for WS-DTT but they have a considerable 10% for total DTT? The authors should indicate a possible explanation for this difference, is this due to**

**the extraction? It is difficult to imagine a sampling difference that will only affect the non-water soluble components.**

**Response:** *This is simply measurement uncertainty. Considering the combined uncertainties from sample preparation and system precision, a 10% bias in $OP^{Total\text{-}DTT}$ is very reasonable. This bias is mainly believed to be due to differences in sampling flow rates of the samplers, which affects the filter loading and has a secondary effect on the cutoff size of the particle collection. Cut size may have some influence since water-soluble and insoluble $OP^{DTT}$ have different size distributions (Fang et al., 2016), the bias in $OP^{WS\text{-}DTT}$ and $OP^{Total\text{-}DTT}$ due to the shift of the cutoff size can be different.*

*Reference:*

Fang, T., Zeng, L., Gao, D., Verma, V., Stefaniak, A. B., and Weber, R. J.: Ambient size distributions and lung deposition of aerosol oxidative potential: a contrast between soluble and insoluble particles, Environ Sci Technol, 2017.

3) **Line 116. Authors indicate that the results from the comparison between extraction by sonication or shaking are similar; although the correlation is reasonable, the scattering is significant. This variation with extraction may indicate an effect of the extraction method depending on the chemical composition of the PM collected and extracted.**

**Response:** *Since linear regression is a statistical model, the error terms in the fitting results do not only result from the effects of measurements, but is also due to some statistical factors, such as sample size and the distribution of the data points. In this test, in addition to the variation with extraction, the small sample size (N=7) and the relatively concentrated data distribution (OP values are mainly in the range of 0.16-0.24 nmol/min/m$^3$) can also lead to the significant scattering. Therefore, we cannot draw any reliable conclusions just based on the error terms. Note that the findings are consistent with those reported by other researchers (see Antinolo, M., Willis, M. D., Zhou, S. M., and Abbatt, J. P. D.: Connecting the oxidation of soot to its redox cycling abilities, Nat Commun, 6, 2015).*

4) **Line 150. Method 1. After extraction with water and resuspension of the extracted solution, there is a considerable amount of sample removed from the filter. These particles are not then extracted with methanol which may lead to underestimating the OP of the methanol extract. Did the authors estimate the loss of particle mass during the first extraction? How much of the original sample remain in the vial for the consecutive extraction? It will be important to know how much is lost if corrections are to be made for more accurate comparison of the OP of the different fractions. Thus, the OP-Total DTT1 (line 164) obtained by adding the OP of each fraction is not accurate and already biased for comparison with the other extraction methods. To obtain a more accurate measurement of the OP of each fraction, and as result the total obtained using this method, the vial containing the extracted sample could be centrifuged and the supernatant removed. This way resuspended particles**

**will remain in the vial for further extraction. Particles can be resuspended by short vortex mixing, and the sample ready for methanol extraction.**

**Response:** *The reviewer's description does not follow the procedure of method 1. The filter is extracted in water and all of the water extract separated from the filter and the liquid sample then filtered with a syringe filter and the DTT levels determined. The original filter is then extracted in methanol… The reviewer is right in that the liquid filtration after water extraction can remove insoluble particles that would have been measured in the methanol-soluble fraction, but now removed from the analysis (ie they won't be measured with the water-soluble or subsequent water-insoluble fraction), which can cause the under-measurement of OP. This is exactly why the method is not preferred and the total $OP^{DTT}$ levels are likely lower compared to the other methods that do not have this limitation. We did not measure this fraction lost since it is likely very small and would be difficult to perform, and furthermore is not routinely done by researchers using this method. The suggested method involving centrifuging could be undertaken, but again, is generally not done because it is too complex. (See first comment above on the assertion that measurements of total $OP^{DTT}$ are straightforward). Finally, the point of this paper is to compare the stated methods of measuring total $OP^{DTT}$.*

5) **Line 177. Method 2. Again, there is a bias in the method as a considerable fraction of the collected PM is removed with the filter.**

   **Response:** *Yes, there can be a bias in method 2, and this is just one reason why it likely is not an ideal approach to measuring total $OP^{DTT}$. Again, the point of this paper is to compare the methods, as they are stated. Note that the bias cannot be large since no large differences were observed between the methods of measuring total $OP^{DTT}$. Because one would then be looking for small differences, (the reviewer gives a possible cause), quantifying the cause of the bias would be very difficult, but again, this is not the point of the paper.*

6) **Depending on the particle size, extraction efficiency varies. What was the extraction efficiency in this case? How much of the sample was removed with the filter? Ultrafine particles are not extracted from the filters easily and as many studies have shown, they are usually the main contributors to the OP of PM. As a result, the measurement is not accurate and comparison with other methods is not valid. Corrections could be made if the efficiency was known and OP measurements adjusted accordingly.**

   **Response:** *We assume that for water-soluble $OP^{DTT}$ measurements, the extraction efficiency is 100% for all sizes of particles. We also agree that extracting particles from a solid surface is difficult due to adhesive forces, and increases in difficulty for smaller particles. Ultrafine particles have significant intrinsic $OP^{DTT}$ (per-PM-mass basis), however, their contributions to air-volume-normalized fine particle OP (used in this study to evaluate human exposure) is small (see Fang et al, 2017). Therefore, the lower extraction efficiency of ultrafine particles is not likely a significant issue. Also this is why method 3 is preferred, where the assay is performed with the filter present so that the*

*chemical reactions can occur on surfaces of particles still attached to the filter. We would like to point out that this comment directly contradicts the first comment made by the reviewer, that measurement of total $OP^{DTT}$ is much easier than water-soluble $OP^{DTT}$.*

***Reference:***

Fang, T., Zeng, L., Gao, D., Verma, V., Stefaniak, A. B., and Weber, R. J.: Ambient size distributions and lung deposition of aerosol oxidative potential: a contrast between soluble and insoluble particles, Environ Sci Technol, 51, 6802-6811, 2017.

7) **Line 192. Automated system. In the description of the analytical method authors indicate that the system lines, valves and pump are only clean after a sample is run, have the authors considered interference in the results by cross contamination and sample interference between consecutive injections of the same sample? The solution remaining in the lines, ports and valves could interfere with the signal of the next injection. As the time between injections is considerable, reaction between DTT and solution can continue while in the lines and valves.**

**Response:** *In the automated system description, we do discuss the cleaning of the system after each sample run. However, this is not the only cleaning step, and there are other cleaning procedures during the analysis. For example, when the addition of each chemical is completed, the syringe pump is washed by pushing DI through. The DI water used for liquid dilution can also clean the tubing. Air is pushed though the tubing to avoid liquid residual. These steps are too detailed to be mentioned in a brief system description. We are providing the program code in the SI so that the reviewer can see more details.*

*We note that the system analysis results are consistent with the results obtained by the manual method which is considered the most accurate (shown Fig. 4) and also considers this cross contamination issue, indicating the accuracy of the system described here.*

8) **Line 305. This sentence is speculation. Unless a comprehensive chemical analysis is conducted to identify the nature of the compounds associated with the sample, the statement is authors speculation.**

**Response:** *Page 11, line 311-312: the sentence has been replaced with "The lower $OP^{Total-DTT-1}$ may be due to liquid filtration after water extraction.".*

9) **Line 311. Based on pervious comments made regarding the methods, this statement may not be correct. There is considerable amount of sample removed from the vial prior to the methanol extraction, which may lead to underestimation of the real potential of the insoluble components. If authors did not quantify the particle loses the methanol extract OP cannot be assumed accurate.**

**Response:** *This has been extensively discussed above. Again, we agree that filtration of water extracts may lead to underestimation of OP levels of sequential methanol extracts.*

*However, the data show this is not a large bias. Furthermore, this will not significantly affect the constituents that can be extracted by methanol. As a consequence, the correlation between $OP^{WI\text{-}DTT\text{-}3}$ and $OP^{sM\text{-}DTT}$ can still provide information about the properties of OP-related water-insoluble species.*

10) **Line 335. The OPsM-DTT is not a direct measure of the oxidation potential of water insoluble component. Components associated with PM present different solubility in different solvents. Many components are not extracted by methanol, and required other organic solvent for extraction. Extraction with hexane, dichloromethane, or acetonitrile, as examples, may result in different OP of the insoluble fraction. Thus, this reviewer disagrees with the statement made by the authors. It will be very difficult to directly measure the OP associated with the insoluble material. The more accurate measurement will be an estimation based on the difference between total OP and water-soluble (filtered) OP.**

   **Response:** *The reviewer is right. On page 12, line 343-345: the sentence has been modified to* "$OP^{WI\text{-}DTT}$ obtained in method 1 is determined from the direct measure of $OP^{sM\text{-}DTT}$, whereas $OP^{WI\text{-}DTT}$ is determined by difference for method 2 and 3, which leads to larger uncertainty and more scatter associated with these data*".

11) **Line 350. A contribution of 35 to 42% of the insoluble material to the total OP is not "some contribution", it is a significant fraction of the total OP.**

   **Response:** *"some contribution" here is indicated by the correlation, not based on the $OP^{WI\text{-}DTT}$-to-$OP^{Total\text{-}DTT}$ ratio.*

   *Page 12, line 360-361: the sentence has been changed to* "The correlation coefficients between $OP^{WI\text{-}DTT}$ and $OP^{Total\text{-}DTT}$ were 0.87 and 0.84 for quartz filters at GT and RS, respectively (Table S1), which reflects the contribution of insoluble species to total OP as well*".

12) **Line 364. This sentence is again speculation. The authors do not present any results regarding the chemical composition to support the idea of secondary compounds as main contributors to the oxidative potential. Identification of primary and secondary compounds, as well as correlation between the different fractions, is required to validate this statement. Wind pattern, local emission sources and measurements at each location are needed to indicate the presence and contribution of primary and secondary compounds.**

   **Line 395. Again, this is speculation by authors. Unless the correlation between primary and secondary compounds with OP is presented, there are no bases to make this statement.**

   **Response:** *In the comments above, the reviewer questions the validation of the statement that PM OP contributors are largely secondary. We would like to address them all at once. Studies have shown that $OP^{DTT}$ is mainly attributed to oxidized aromatic species*

*and soluble transition metals. Vehicles emit precursors for both: tailpipe emissions from incomplete combustion are sources of PAHs, the precursor for quinones; and brake or tire wear and road dust are precursor for soluble transition metals. While the primary emissions exist, which is demonstrated by the much higher level of EC and metals at RS, the homogenous distributions of $OP^{DTT}$ suggest secondary atmospheric processing plays an important role in converting the precursors to DTT-active species. This is supported by a large and growing body of research.*

13) **Line 394. OP measurements for two points, GT and RS, are not representative of a wide area, so concluding the measured OP were largely spatially homogeneous is too broad of a statement.**

    **Response:** *Page 14, line 405: deleted "largely".*

14) **Line 520. Table 2. Authors present correlation between OP measured under different methods and chemical speciation. Among the chemical compounds presented by the authors, K shows important correlation with OP. However, this compound has not been shown to be redox active or a chemical that can react with DTT. Why do the authors include this compound in their list and their correlations? If this is not a redox compound it does not contribute to the OP, and the correlation is not significant; unless the compound is used as a marker for sources that can contribute to other compounds that do contribute to the OP of the sample. A coherent explanation is needed regarding the inclusion of K in the table and results.**

    **Response:** *K was presented as a marker of biomass burning. Based on the reviewer's suggestion, we have modified the manuscript to include this information.*

    *Page 8, line 228-229: "The elements of interest included species that possibly play a role in ROS generation (Fe, Mn, Cu; Schoonen et al., 2006) and K, a marker of biomass burning (Artaxo et al., 1994)."*

**Response to anonymous referee #2 comments:**

**I am fine with the paper, I do not have any other comments, I recommend acceptance.**

**Response:** *We thank the review for their recommendation on the acceptance.*

**Response to anonymous referee #3 comments:**

This article reported three methods of analyzing aerosol oxidative potential (OP) based on DTT assay, focusing on different sample extraction strategies to distinguish soluble components from the total aerosol particulate matter (PM). I think this is a good work with practical meaning to assist the understanding of oxidative aerosol's transport in atmosphere. I have one major suggestion as below.

1) Figure 1 delivered a clear demonstration of the three sample extraction and analysis methods. However, I think a better job can be done to explain what component(s) from the total PM species is measured by each method throughout the text. Basically, two factors, including extraction solvent (water versus methanol) and filter (i.e., with or without filtering) differentiate the three methods. Which component in the aerosol is screened by each factor should be clearly specified to elucidate what is the actual difference across the three methods. From my understanding, the use of water or methanol discriminates the water-soluble (herein "soluble" matter includes dissolved molecules/ions and "dispersed" solids (or small particles)) and methanol-soluble component, whereas filtering or not filtering a sample discriminates dissolved molecules/ions and dispersed solids in that sample. Overall, extraction combined with these two factors actually categorizes the total PM into six groups: water-insoluble species (those which cannot be extracted into water), water-soluble molecules/ions, water-soluble solids (dispersible small particles), methanol-insoluble species (those which cannot be extracted into methanol), methanol-soluble molecules/ions, and methanol-soluble solids. The authors should clarify which groups are measured in each of the three methods, either with a diagram or plain text.

   **Response:** *We thank the reviewer for an interesting suggestion. However, we feel that the detailed presentation approach the reviewer suggest is so detailed that it may hinder understanding the contrasts in the methods. Furthermore, focusing on solubility is not the complete issue, there are the extraction issue (how solid particles are removed from the filter media) and transport issue (how solid particles are transported to the location where DTT analysis is performed). Solubility is related to this, but we feel this moves the focus away from the physical aspects involved in making a total $OP^{DTT}$ measurement. Basically, we are trying to include the contribution of water-insoluble species to $OP^{DTT}$ in two ways: one is to get the water-insoluble components dissolved in organic solvent (methanol was chosen in this study as it is widely used), and the other is to get solid particles exposed to the DTT assay (i.e., don't dissolve them). To be clear, we clarify which species is measured for the three methods in the summary table (Table XXX).*

**Elucidating the above may rationalize a few ambiguous places better in the articles. What follows are some examples.**

2) **L59-60,76-77, there seem to be some conflicts between these two references: the first place says up to 99% of the DEP CANNOT be extracted by water or methanol while the second one says the measurement is based on water extraction. "after which the filter was removed from the analysis" is very confusing. What are the authors trying to say here?**

Response: *There is no contradiction between the two citations which refer to different things. The sentence in line 76-77, refers to the method used to measure the **total** OP of diesel exhaust particles (DEP). The method is based on water extraction, but insoluble particles remain suspended in the extracts. In McWhinney et al., the OP of water and organic extractable fractions were also determined. The difference between total and extractable fraction is what we are referring to in line 59-60, i.e., the **non-extractable** fraction.*

*We have modified the sentence on page 3, line 76:* "McWhinney et al. (2013) measured total redox activity of DEP using particle suspensions that were obtained by a water-extraction procedure with the filter removed after extraction.".

3) **L100-101: what is "10%" here?**

Response: *Page 4, line 99-101: the sentence has been changed to* "The bias between the two High Volume samplers was assessed by running them side-by-side at GT for 9 days. The measurements were within 10% for both water-soluble $OP^{DTT}$ ($OP^{WS-DTT}$) and total $OP^{DTT}$ ($OP^{Total-DTT}$) (obtained by method 3, described below) (Supplement Fig. S2).".

4) **L300-323: The justification of choosing method 3 over the other two is poor. It may be improved if the authors can specify each species category corresponding to each separation technique as I suggested. I have specific questions as below.**

Response: *In line 300-323, the methods are compared only based on the levels of OP measurements. The conclusions here are not an overall evaluation. The specific changes in the manuscript are given in the response below.*

5) **L305-309, it seems method 1 is more consistent with SCAPE study, than method 3; then why is method 3 selected over method 1?**

Response: *Only method 1 was deployed during the SCAPE study, so our data was not compared with the SCAPE data for method 3. The purpose of comparing the data in this*

*study to SCAPE data is to confirm that our data are reasonable and fall into the normal range of ambient samples.*

*To clarify, the sentence on page 11, line 312-314 has been changed to "The ratios of $OP^{sM-DTT}$ to $OP^{WS-DTT}$ are $0.34 \pm 0.14$ (N=35) at GT and $0.37 \pm 0.12$ (N=31) at RS, which are consistent with the ratios from SCAPE data ($0.27 \pm 0.08$, N=198; unpublished data) and fall into the typical range of ambient samples.".*

6) **L309, what is indicated by "very little correlation between $OP^{WI-DTT-3}$ and $OP^{WS-DTT}$?**

**Response:** *Line 316-318: "There was very little correlation between the $OP^{WI-DTT-3}$ and $OP^{WS-DTT}$ with Pearson correlations of r = -0.23 and -0.51 at GT and RS sites, respectively (see Supplement Table S1), which further indicates the importance of water-insoluble compounds to a total OP measurement.".*

7) **L315, I am uncomfortable with "overall" here, because the only justification of method 3 over method 1 is that it measures higher total-DTT (which is actually quite common sense). The whole paragraph seems quite over-informative and little relevant to the conclusion.**

**Response:** *We have modified the expression of the conclusion and rearranged some sentences to clarify the logic.*

*Page11, line 305-323: "Figure 5 shows the $OP^{DTT}$ $m^{-3}$ comparison between method 1 and 3 at both GT and RS sites. In general, the response of the DTT assay of method 3 was significantly higher than that of method 1 at the 95 % confidence level (paired t-test: p = 0.028 at GT, N=35; p<0.001 at RS, N=31). The results are expected since in method 1, both the water and methanol liquid extracts are filtered, potentially removing species that could have been DTT-active but remained attached to solid particles. A few observations where $OP^{Total-DTT-3}$ is less than $OP^{Total-DTT-1}$ are likely due to propagation of errors for the summation method (method 1) combined with variability in the extraction process for each method. The mean $OP^{Total-DTT-1}$ to $OP^{Total-DTT-3}$ ratio at GT was close to 1 (ratio = 0.95) and also higher than that at RS (ratio = 0.85). This may imply that method 1 can be more effective for extracting aged PM species. The ratios of $OP^{sM-DTT}$ to $OP^{WS-DTT}$ are $0.34 \pm 0.14$ (N=35) at GT and $0.37 \pm 0.12$ (N=31) at RS, which are consistent with the ratios from SCAPE data ($0.27 \pm 0.08$, N=198; unpublished data) and fall into the typical range of ambient samples. The water-insoluble OP determined by the difference in $OP^{Total-DTT-3}$ (which includes solid particles) and $OP^{WS-DTT}$ ($OP^{WI-DTT-3} = OP^{Total-DTT-3}$ - $OP^{WS-DTT}$) to $OP^{WS-DTT}$ ratio, by contrast, is $0.45 \pm 0.25$ at GT (N=35) and $0.67 \pm 0.35$ at RS (N=31). There was very little correlation between the $OP^{WI-DTT-3}$ and $OP^{WS-DTT}$ with Pearson correlations of r = -0.23 and -0.51 at GT and RS sites, respectively (see Supplement Table S1), which further proves the importance of water-insoluble compounds in OP assessment. Additionally, $OP^{WI-DTT-3}$ was weakly correlated with $OP^{sM-}$*

$^{DTT}$ (Pearson correlation: r = 0.31 at GT; r = 0.04 at RS). Based on these data, it is clear that there were species associated with water-insoluble OP$^{DTT}$ not extracted by methanol and that remain attached to solid particles. This analysis shows that filtering the liquid extract, even if methanol solvent is used, will result in a substantial underestimation of OP$^{Total\text{-}DTT}$. Therefore, in terms of the OP response, method 3 is preferred to method 1. Furthermore, the comparison between these two methods can provide insights into the water-insoluble components that contribute to PM OP."

8) **L319-323, the same problem is with the comparison of method 2 and method 3. There is not actual justification for why method 3 is chosen over method 2. Information is very redundant and little relevant to the conclusion.**

   Response: *As we replied in question 4), the comparison in this paragraph was only based on OP response.*

9) **L330-339, it's somewhat ironic here. I was told that method 1 is better than method 2 and 3 in terms of seeking correlation between OP-DTT and PM compositions but then suddenly method 3 was selected to be used... Why?**

   Response: *We have modified this section to clarify the logic.*

   *Line 333-345:* "A correlation analysis was performed between measured PM$_{2.5}$ chemical constituents and OP$^{DTT}$ determined by the three methods. Figure 7 shows the correlation results (detailed values are provided in Table S2). It is seen that OP$^{Total\text{-}DTT\text{-}3}$ is better correlated with the measured species than OP$^{Total\text{-}DTT\text{-}1}$ and OP$^{Total\text{-}DTT\text{-}2}$. Compared with OP$^{WS\text{-}DTT}$, the stronger correlations between OP$^{Total\text{-}DTT\text{-}1}$ and PM species suggests that OP$^{Total\text{-}DTT\text{-}1}$ captures more chemical components contributing to DTT than OP$^{WS\text{-}DTT}$. In contrast, OP$^{Total\text{-}DTT\text{-}2}$ is correlated with the least number of measured PM species.

   By subtracting OP$^{WS\text{-}DTT}$ from OP$^{Total\text{-}DTT}$, OP$^{WI\text{-}DTT}$ is determined for the three methods. In general, the correlations between OP$^{WI\text{-}DTT}$ and PM species are mediocre for all three methods, with a slightly better performance of method 1. The water-insoluble OP$^{DTT}$ determined by method 1, i.e. OP$^{sM\text{-}DTT}$, has good correlation with OC at GT and OC, EC and water-soluble Fe at RS. Verma et al. (2012) also showed good correlations between OP$^{DTT}$ of filtered methanol extracts and OC and EC, and attributed this association to water-insoluble organic carbon species (WIOC) that dissolve in methanol. Thus, OP$^{sM\text{-}DTT}$ in method 1 is likely attributed to some fraction of the WIOC. OP$^{WI\text{-}DTT}$ obtained in method 1 is determined from the direct measure of OP$^{sM\text{-}DTT}$, whereas OP$^{WI\text{-}DTT}$ is determined by difference for method 2 and 3, which leads to larger uncertainty and more scatter associated with these data."

10) **L348: I recommend clarifying how 35-42% is determined and specify the error range as well.**

**Response:** *The manuscript has been modified.*

*Page 12, line 357-360: "The ratios of $OP^{WS-DTT}$ to $OP^{Total-DTT}$ were on average $65 \pm 10$ % (insoluble accounts for $35 \pm 10$ %) and $65 \pm 14$ % at GT, compared to $62 \pm 12$ % and $58 \pm 10$ % at RS, for quartz and Teflon PM samples, respectively. Thus, $OP^{Total-DTT}$ of $PM_{2.5}$ contained on average 35 to 42 % insoluble species."*

11) **Overall, I strongly suggest the authors give their recommendations on which method should be used in what scenarios or for what purposes.**

   **Response:** *We added a summary table (Table 2) and modified the text accordingly.*

   *Page 12, line 346-350: "The overall assessment of the three methods is summarized in Table 2. By comparison, it is found that method 3 has better precision, more comprehensive response (i.e., generally highest $OP^{Total-DTT}$), stronger correlations with PM components, and easiest filter preparation (extraction) process, all of which provide an efficient way for $OP^{Total-DTT}$ determination. The other two methods have some value owing to their insights into the attributes of water-insoluble OP contributors. In a subsequent study, discussed next, only Method 3 was utilized to measure $OP^{Total-DTT}$ of PM for Teflon filters.".*

**Other minor suggestions:**

12) **L133-134: provide possible explanations on what gives the response in blank samples.**

   **Response:** *Based on the reviewer's suggestion, the manuscript has been modified to include the possible explanations for blank OP values.*

   *Page 5, line 134-136: "Since DTT is a relatively unstable compound, it can react with dissolved oxygen in the liquid in the absence of particles (Kumagai et al., 2002), resulting in OP response in blanks. The blank OP values are also due to trace levels of contaminants on the filter, in the DI-water, and introduced during sample preparation."*

13) **L275-278: Maybe provide a figure in SI to illustrate the validation with five ambient samples as well?**

   **Response:** *The figure (Fig. S7) is added in Supporting info.*

14) **Table 1: Does "N" filters correspond to different samples? (I assume CV is determined with three replicates on each filter (each sample)?) and what is the range standing for.**

**Response:** *Yes, "N" stands for the number of filters used for duplicate measurements. CV is determined with triplicate on each filter. CV values for each method vary from sample to sample, and the CV range is given based on the maximum and minimum measured.*

*To be clearer, change "Coefficient of variation (CV)" in Table 1 to "Coefficient of variation (CV)* from triplicate*".*

15) **Table 2: I think a figure is better to let readers see the correlations of different variables, although I know there are many comparisons here. Maybe the authors can think if the illustration here can be improved.**

**Response:** *We added polar plots (Fig. 7) showing the correlation coefficients between $OP^{DTT}$ and different PM species. The table is moved to Supplement (Table S2).*

**Response to anonymous referee #4 comments:**

**Review for A method for measuring total aerosol oxidative potential (OP) with the dithiothreitol (DTT) assay and comparisons between an urban and roadside site of water-soluble and total OP by Dong Gao et al.**

**This manuscript describes a careful study further developing a technique previously published by the group. Different aerosol sampling and extraction methods are compared and first field data are presented to quantify aerosol bound OP. After addressing the points listed below, I recommend publication in AMT.**

**Response:** *We thank the reviewer for the recommendation on acceptance. Our responses and corresponding changes made in the manuscript (highlighted in red) are given below. Note that the comments the reviewer gave were based on the initially submitted manuscript, so the line numbers in the responses may be inconsistent with that in the reviewer comments.*

1) **p.2, line 37: OP(total-DTT) is that the unfiltered methanol extract fraction? It might be best not to use undefined abbreviations in the abstract.**

   **Response:** *Thank the reviewer for the suggestion. To be clear, we added* "Therefore, the third method was applied to the field study for the determination of total OP." *on page 1, line 17-18.*

2) **p. 2, line 40-41: This sentence is not clear to me. Was the same DTT analysis performed on Teflon filters? Please be more specific.**

   **Response:** *The sentence has been modified.*

*Page 1, line 22-23: "Same DTT analyses were performed and similar results were found using particle composition monitors (flow rate of 16.7 L min$^{-1}$) with Teflon filters."*

3) **p. 3, line 62/63: It would be good to mention also a reference for the DCFH assay besides DTT and ascorbic acid.**

**Response:** *Reference for the DCFH assay has been added.*

*Page 2, line 39-40: "Various methods have been developed to assess PM OP (Ayres et al., 2008; Cho et al., 2005; King and Weber, 2013; Mudway et al., 2004; Shi et al., 2003; Wang et al., 2011)."*

4) **p.8, line 167 – 169. I do not agree with the statement that sonication has no effect on radical formation. The paper cited, Miljevic demonstrates explicitly the opposite. Fig. 3 in that paper shows clearly that sonication strongly oxidises DTT! It is not clear to what section in the SI this sentence refers to. Fig. S3 or S5? In Fig S5 both axes are labelled that same. So it is not clear what Fig S5 is showing. This statement has to be worded much more carefully and the results presented and the potential effects of sonication have to discussed critically by representing literature results correctly!**

**Response:** *We agree that Miljevic did report the sonication affects DTT, we, and others, however, have failed to see such an effect. We found that the DTT analysis for samples extracted by sonication and shaking produced similar results, indicating that any radicals formed during sonication caused negligible bias in the measurement of PM OP. The work of Antinolo et al. (2015) shows the same results.*

*To be clearer, the sentence has been modified. Page 4, line 116-118: "Considering the potential for radical formation during the sonication process (Miljevic et al., 2014), experiments using sonication versus shaking were done. Little difference observed in OP for sonication versus shaking OP indicated negligible bias in OP$^{WS-DTT}$ measurement due to sonication, see Supplementary Material Fig. S3."*

*In terms of SI Fig. S5, we have corrected the graph in the updated manuscript.*

   **Response:** *Some OP$^{DTT}$ values have similar correlations with both water-soluble and total elements, which is possibly due to the good correlation between water-soluble metal and total metal concentrations (see Fig. S1). However, we can still observe that OP$^{Total\text{-}DTT\text{-}3}$ is correlated better with total metal than with water-soluble metal for both GT and RS site. This may imply that the total OP measurements can capture some water-insoluble species, and therefore strengthen the relationship between OP and total elements.*

[revised manuscript text omitted]